# Energy-Based Transformers are Scalable Learners and Thinkers

**Alexi Gladstone[1,2], Ganesh Nanduru[1], Md Mofijul Islam[1,3], Peixuan Han[2],**
**Hyeonjeong Ha[2], Aman Chadha[3,4], Yilun Du[5], Heng Ji[2], Jundong Li[1], Tariq Iqbal[1]**

[1]UVA   [2]UIUC   [3]Amazon GenAI[†]   [4]Stanford University   [5]Harvard University

🌐 energy-based-transformers.github.io    ⭘ github.com/alexiglad/EBT

## Abstract

Inference-time computation, analogous to human System 2 Thinking, has recently become popular for improving model performance. However, most existing approaches suffer from several limitations: they are modality-specific (e.g., working only in text), problem-specific (e.g., verifiable domains like math and coding), or require additional supervision/training on top of unsupervised pretraining (e.g., verifiers or verifiable rewards). In this paper, we ask the question *"Is it possible to **generalize** these System 2 Thinking approaches, and **develop models that learn to think solely from unsupervised learning?"*** We find the answer is **yes**, by learning to explicitly **verify** the compatibility between inputs and candidate-predictions, and then re-framing prediction problems as optimization with respect to this verifier. Specifically, we train Energy-Based Transformers (EBTs)—a new class of Energy-Based Models (EBMs)—to assign an **energy** value to every input and candidate-prediction, enabling predictions through energy minimization until convergence. To support this approach, we introduce several key techniques for stable and parallelizable training, which enable the emergence of strong System 2 Thinking capabilities and scalable EBMs. Across discrete and continuous modalities, we find EBTs outperform the Transformer++ approach, scaling up to 35% faster during pretraining, and improving inference-time performance by up to 29%. EBTs also surpass Diffusion Transformers on image denoising while requiring 99% fewer forward passes. Moreover, System 2 Thinking with EBTs yields larger performance gains on data that is farther out-of-distribution, and EBTs achieve better results than existing models on most downstream tasks despite achieving the same or worse pretraining performance, enabling EBTs to generalize better than existing approaches. Consequently, EBTs are a flexible and promising new approach for scaling both the **learning** and **thinking** capabilities of models.

## 1 Introduction

In psychology, human thinking is often classified into two different types: System 1 (thinking fast) and System 2 (thinking slow) Evans (2011); Frankish (2010); Kahneman (2011); Kahneman et al. (2002). System 1 thinking is characterized by quick, intuitive and automatic responses, relying on previous experience to solve simple or familiar problems. Alternatively, System 2 Thinking is slow and deliberate, requiring effort to solve complex problems that go beyond automatic pattern recognition, such as in mathematics or out-of-distribution situations Goel et al. (2000); Neys (2006). Current models perform well on tasks suitable for System 1 thinking Li et al. (2025c), but continue to struggle with tasks that demand System 2 capabilities Mirzadeh et al. (2024); Yan et al. (2025).

As a result, System 2 Thinking has become a growing research focus, driving the development of foundation models such as O1 Jaech et al. (2024), R1 Guo et al. (2025), and Claude Anthropic (2025). These "reasoning models" excel on math and coding benchmarks by increasing the time models

---

Correspondence to Alexi Gladstone: ✉ alexigladstone@gmail.com. [†]Work does not relate to position at Amazon.

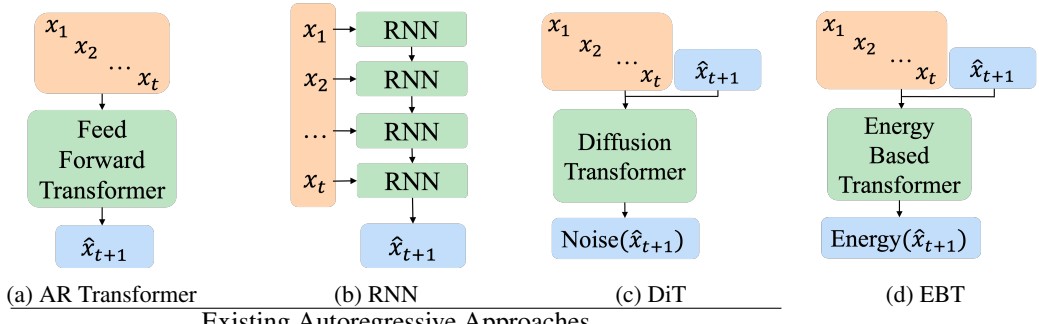

|        (a) AR Transformer        |        (b) RNN        |        (c) DiT        |        (d) EBT        |

Existing Autoregressive Approaches

Figure 1: **Autoregressive Architecture Comparison.** (a) Autoregressive (AR) Transformer is the most common, with (b) RNNs becoming more popular recently Gu & Dao (2023); Peng et al. (2023). (c) Diffusion Transformers (DiTs) Li et al. (2025b); Peebles & Xie (2023) are similar to EBT, being able to dynamically allocate computation during inference. However, diffusion models are not trained as explicit verifiers, unlike EBTs.

spend thinking. However, publicly available information from the open-source R1 model Guo et al. (2025), suggests that the Reinforcement Learning (RL) approach for training these models only works in domains where rule-based rewards can easily verify answers, such as math and coding. This limits applicability, and often harms performance on tasks such as writing OpenAI (2024); Su et al. (2025). Moreover, recent evidence indicates this approach may not foster new reasoning patterns, restricting performance on tasks requiring exploration Yue et al. (2025). For a discussion on Related Works, see Section E.

As one of the primary goals of AI is to figure out how we can create systems that **learn to think on their own** on any problem type, these approaches ultimately bring about the following core research question: "*Can we **rely entirely** on **unsupervised learning** to develop **System 2 Thinking?***" Such a capability could enable *generalization* of current System 2 Thinking approaches to *many problems*, *many modalities*, and *avoid the reliance on human, reward, or model supervision*.

We argue and our empirical results suggest that the answer to this question is **yes**, but that there are several limitations in existing models that prevent this general Thinking from emerging. Particularly, when comparing the qualities of human System 2 Thinking with current modeling approaches (Figure 1, Table 1), we observe several key differences, outlined below as two key Facets of System 2 Thinking:

**Facet 1: Dynamic Allocation of Computation.** Humans naturally allocate varying amounts of effort to different tasks depending on difficulty, which is widely supported by psychology and neuroscience Ditterich (2006); Kahneman (2011); Rougier et al. (2005).[2] For example, a decision regarding whether to change careers generally takes more time than deciding what to eat.

**Facet 2: Verification of Predictions.** In addition to allocating computation, human thinking also benefits from the ability to verify predictions Alkouri (2016);

Table 1: **Architectures and Cognitive Facets.** For each prediction, Feed-Forward (FF) Transformers and RNNs generally[1] have a finite amount of computation. DiTs (Diffusion Transformers) can increase inference computation by denoising longer, but lack explicit prediction verification. In contrast, EBMs support dynamic computation through flexible iteration, and give an energy scalar for prediction verification.

| Arch. | Dynamic Compute Allocation (Facet 1) | Prediction Verification (Facet 2) |
|---|:---:|:---:|
| FF Trans. | ✗ | ✗ |
| RNNs | ✗ | ✗ |
| DiTs | ✓ | ✗ |
| EBTs | ✓ | ✓ |

Loesche et al. (2018), which can guide decisions about when to stop thinking or to select the most accurate predictions. This also supports more dynamic inference time behavior, such as early stopping when a prediction is known to be correct, or allocating more compute when a problem is difficult. For more information on additional Facets, please refer to Section F.

---

[1]Recent works attempt to enable dynamic computation per prediction Geiping et al. (2025); Hao et al. (2024), but these approaches are generally not modality agnostic and have not been widely adopted.

[2]We refer to this dynamic computation at the granularity of **each** prediction, meaning current LLMs built with AR transformers/RNNs cannot dynamically allocate compute per token. See Section G for more info.

Figure 2: **EBT for Autoregressive Modeling.** Each blue box corresponds to a different prediction at each step of the thinking process, where the initial prediction starts as random. At each step, a new prediction is fed into the model, which gives an energy scalar for the prediction's current *compatibility* (unnormalized likelihood) with the context (Facet 2). Then, the gradient of this energy with respect to the prediction is calculated and used to update the prediction. This gradient descent update is done iteratively to refine the prediction until convergence of the predicted energy, which allows for dynamic use of computation (Facet 1).

To achieve the two facets described, we propose viewing thinking as optimization with respect to a learned verifier, which evaluates the compatibility (unnormalized probability) between an input and candidate prediction (Figure 2). Specifically, we train Energy-Based Models (EBMs) to learn an energy (unnormalized probability) landscape over all possible input-prediction pairs, where lower energy indicates higher compatibility (Facet 2). Thinking then corresponds to starting from an initial random prediction and refining it through optimization along the energy landscape until convergence (visualized in Figure 3). This naturally enables dynamic compute allocation (Facet 1) in the form of more challenging problems utilizing additional optimization steps.

While this thinking perspective is promising, Energy-Based Models (EBMs) have struggled with scalability Du & Mordatch (2019), with no known foundation EBMs. This stems from issues with training instability and long training times Arbel et al. (2020); Du & Mordatch (2019). To address these challenges, we introduce Energy-Based Transformers (EBTs), or Transformers specifically for EBMs. We further propose practical training improvements, theoretical insights into EBM training scalability, and novel *energy landscape regularization* techniques that improve System 2 Thinking.

To assess learning and thinking scalability, we compare EBTs to the Transformer++ (autoregressive) and DiT (bidirectional) across discrete and continuous modalities. EBTs show up to 35% higher scaling rates than the Transformer++ across data, batch size, parameters, FLOPs, and depth. At inference, EBTs outperform existing models on System 2 Thinking—for example, by improving language model performance by 29% more than the Transformer++, and by outperforming DiTs in image denoising with 99% fewer forward passes. We observe two key effects: (1) EBTs often outperform baselines at inference even with worse pretraining performance, demonstrating the importance of System 2 Thinking; and (2) System 2 Thinking yields greater gains on more out-of-distribution data, paralleling human thinking. We believe the EBT implementations, along with novel techniques for EBMs to maximize the learning and thinking scalability, will advance the EBM approach by addressing key challenges in stable, parallelizable, and efficient training.

## 2 ENERGY-BASED TRANSFORMERS (EBT) INTUITION

### 2.1 LEARNING TO VERIFY

Verifying solutions is often substantially more tractable than generating them, a distinction well-known in complexity theory Cook (2023); Goldwasser et al. (2019); Gödel (1956). For example, in solving a maze, verifying the correctness of a given path is significantly easier than discovering such a path. This asymmetry has been recognized and utilized for several decades, notably in the field of cryptography Goldwasser et al. (2019); Lavin et al. (2024); Rivest et al. (1978). EBMs are built on this principle that *verification is easier than generation*: rather than learning to generate directly, as in most existing approaches, EBMs learn to generate by optimizing predictions with respect to the learned energy function (shown in Figure 3).

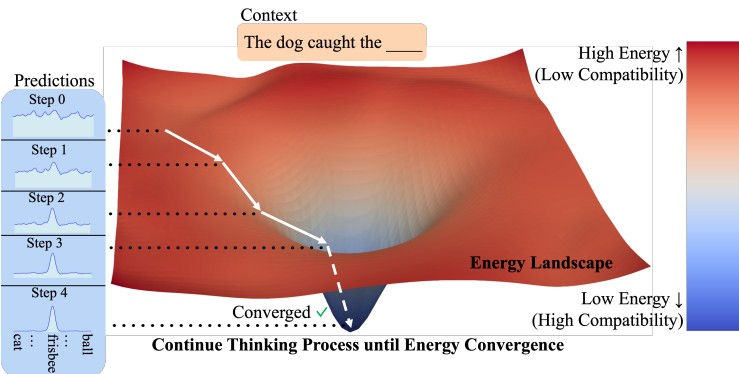

Figure 3: **Thinking Process Visualization.** An example energy landscape and its optimization through gradient descent, interpreted as thinking. Here, the model predicts a distribution over text tokens, progressively shifting from an initial random distribution to the target distribution. At each step, the EBM assigns an energy scalar indicating how **compatible** the current prediction is with the context, visualized as the landscape's height (Facet 2). This scalar's convergence allows the model to determine whether the prediction is adequate or if further thinking is necessary (Facet 1). We include a more detailed toy example in Section C.3. Adapted from Li et al. (2018).

Recent works have attempted to leverage verifiers Ma et al. (2025); Team (2023); Yao et al. (2023), but these approaches decouple the verifier and generator, resulting in adversarial dynamics Ma et al. (2025) and challenges in scalability Yao et al. (2023). For example, researchers combining tree search and LLMs required thousands or even millions of samples to achieve optimal performance Team (2023). In contrast, EBMs **combine** the verifier and generator into a **single model**, where the generator is defined implicitly by the gradient of the verifier Du & Mordatch (2019). We show that this coupling addresses scalability and adversarial issues (Figures 7b and B.4a).

An additional advantage of verifiers is generalization. Because verification is usually easier than generation Swamy et al. (2025), *prediction verification on Out-Of-Distribution (OOD) data is often easier than explicit prediction generation for OOD data* Du et al. (2022). This characteristic often results in better generalization of verifiers than explicit generators Du et al. (2022). This may explain why EBMs often generalize better than existing models Du et al. (2022; 2024), which we further support in our experiments (Figure 7a and Table 4).

### 2.2 LEARNING TO UNDERSTAND

This verifier-centric perspective also relates to a deeper limitation in generative models, referred to as "The Generative AI Paradox West et al. (2023)." Although current generative models achieve strong generative capabilities, they frequently lack basic discrimination skills, such as the ability to assess the plausibility or coherence of their own predictions Stojnić et al. (2023); West et al. (2023), impeding their ability to engage in reasoning, planning, and decision-making Kambhampati et al. (2024); Yan et al. (2025). In contrast, EBMs offer a potential solution to this challenge: as EBMs generate by learning a verifier (which is similar to a discriminator), they develop strong discrimination skills Wang et al. (2023). Experimental results further support this observation (Table 4).

## 3 ENERGY-BASED TRANSFORMERS (EBT) APPROACH

### 3.1 ENERGY-BASED MODELS (EBM) BACKGROUND

Energy-Based Models (EBMs) assign a scalar energy value to each configuration of input variables, enabling them to model the **compatibility** and **interactions** between variables, such as between a context and candidate-prediction. For probabilistic EBMs, this defines a probability distribution using a Boltzmann distribution $p_\theta(x) = \frac{e^{-E_\theta(x)}}{Z(\theta)}$ where $Z(\theta) = \int e^{-E_\theta(x)} dx$ is the intractable partition function involving an integral over all possible values of $x$. To avoid the intractability of the partition function, it is common to work with **unnormalized EBMs**, which dispense of the partition function in favor of representing relative unnormalized probabilities. This formulation shifts the

| **Algorithm 1:** Training | **Algorithm 2:** Inference with Verification |
|---|---|
| **Inputs:** Context $x$, Target $y$, EBM $E_\theta(x, \hat{y})$ | **Inputs:** Context $x$, EBM $E_\theta(x, \hat{y})$ |
| **Hparams:** Steps $M$, Step Size $\alpha$, Loss $J(\cdot)$ | **Hparams:** Steps $M$, Step Size $\alpha$, Samples $N$ |
| 1 Sample $\hat{y}_0 \sim \mathcal{N}(0, I)$; | 1 **for** $j = 1, \ldots, N$ **do** |
| 2 **for** $i = 0, \ldots, M - 1$ **do** | 2 $\quad$ Sample $\hat{y}_{0,j} \sim \mathcal{N}(0, I)$; |
| 3 $\quad \hat{y}_{i+1} \leftarrow \hat{y}_i - \alpha \nabla_{\hat{y}_i} E_\theta(x, \hat{y}_i)$; | 3 $\quad$ **for** $i = 0, \ldots, M - 1$ **do** |
| 4 $\mathcal{L} \leftarrow J(\hat{y}_M, y)$; | 4 $\quad\quad \hat{y}_{i+1,j} \leftarrow \hat{y}_{i,j} - \alpha \nabla_{\hat{y}_{i,j}} E_\theta(x, \hat{y}_{i,j})$; |
| 5 **return** $\mathcal{L}$, *update* $E_\theta$; | 5 **return** $\hat{y}^* = \text{argmin}_j E_\theta(x, \hat{y}_{M,j})$; |

focus from addressing the partition function, to simply assigning low energy to the true data manifold and high energy elsewhere Dawid & LeCun (2024); Du & Mordatch (2019), offering benefits such as scalability to spaces where the true data manifold is thin and therefore a probabilistic EBMs would have an infinite score Dawid & LeCun (2024). In supervised or predictive self-supervised learning (e.g., classification, autoregressive modeling, masked modeling), unnormalized EBMs can be formulated as: $p_\theta(x, \hat{y}) \propto e^{-E_\theta(x, \hat{y})}$, where the goal of the EBM is to learn to predict $\hat{y}$ given $x$.[3]

## 3.2 Scalable EBM Learning

While EBMs offer a flexible modeling framework, training them scalably remains an open research problem. Two primary training approaches exist—contrastive and regularized methods LeCun (2022). Contrastive methods increase the energy of negative samples while decreasing the energy of positive samples. Due to the curse of dimensionality Dawid & LeCun (2024), where the volume of spaces grows exponentially with their dimension, contrastive methods struggle to scale because they must increase the energy of an exponentially higher number of negative samples.

An alternative is to frame EBM learning as an optimization problem Du et al. (2022); Wang et al. (2023), which avoids the curse of dimensionality by implicitly regularizing the energy landscape, enabling scalable learning. In this approach, EBMs are trained to optimize an initial prediction to the ground truth solution through gradient descent, as shown in Figure 3. This pushes the energy landscape to have a local minima surrounding the ground truth solution, thereby regularizing the energy landscape to only have low energy on the true data manifold. Intuitively, this optimization-based training approach is similar to GANs Goodfellow et al. (2014). During the forward pass, EBMs can be seen as a GAN discriminator by giving an energy "verification"; on the backward pass they can be seen a GAN generator by optimizing predictions through energy minimization to try and fool the discriminator.

Training EBMs to perform optimization can be formalized as follows. We begin with an EBM $E_\theta$, an initial prediction $\hat{y}_0$, an input (context) for the model $x$, and seek to predict $y$. We aim to find the minimum energy (most compatible) $\hat{y}$ given an $x$, which we search for using gradient descent:

$$\hat{y}_{i+1} = \hat{y}_i - \alpha \nabla_{\hat{y}_i} E_\theta(x, \hat{y}_i), \tag{1}$$

where $\alpha$ is the step size (formalized in Algorithm 1). Then, the loss can be computed using any standard objective function. Importantly, this loss is backpropagated through the entire optimization process, requiring second-order derivatives (i.e., gradients of gradients). These are computed efficiently via Hessian-vector products, which scale linearly with model size Dagréou et al. (2024), similar to standard gradient descent in feed-forward models. More details and pseudocode can be found in Section I.2.

## 3.3 Scalable EBM Thinking

While this training approach is scalable, achieving smooth energy landscapes with a single local minimum remains challenging on real-world problems. Because $y$ is high-dimensional, the energy landscape spans a high-dimensional space, and must remain well-shaped throughout. To address this, we found three key *energy landscape regularization* techniques to be helpful for learning smoother, more well-behaved energy landscapes, enabling strong thinking capabilities to emerge during training.

First, we found a replay buffer helps simulate longer optimization trajectories, enabling energy landscapes to be well defined near their minimum. Second, a variant of Langevin Dynamics Du &

---

[3]In self-supervised learning, $x$ is some unmasked portion of the original $x$ and $\hat{y}$ is the masked portion.

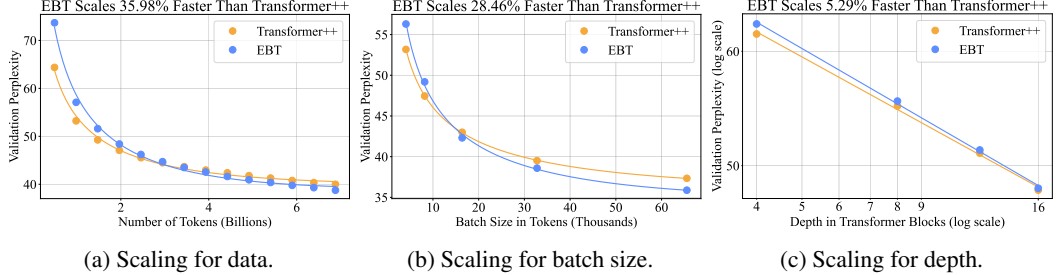

Figure 4: **Language Learning Scalability—Data, Batch Size, and Depth.** A comparison between the scaling of the Transformer++ recipe Touvron et al. (2023) and EBTs across data, batch size, and depth during pretraining. On all axes, EBTs out-scale the Transformer++ recipe significantly, indicating improved data efficiency. The improved depth scaling offers promise for reasoning, where depth is crucial Ye et al. (2024). These results suggest that EBTs offer promise at large data scale.

Mordatch (2019) was found to be helpful for encouraging **exploration** of the energy landscape:

$$\hat{y}_{i+1} = \hat{y}_i - \alpha \nabla_{\hat{y}_i} E_\theta(x, \hat{y}_i) + \eta_i, \quad \eta_i \sim \mathcal{N}(0, \sigma), \tag{2}$$

where $\sigma$ is the magnitude of the noise $\eta$. Without this noise term, exploration is often limited to paths leading directly to the energy minimum, leaving other regions poorly defined. Third, varying the paths taken towards predicting solutions, by randomizing the gradient descent step size $\alpha$ and number of optimization steps, significantly improved generalization. Together, these techniques improved the System 2 Thinking capabilities of models, as confirmed by ablation experiments in Table 2.

With these techniques established, we explored two main Thinking approaches. First, corresponding to dynamic computation allocation (Facet 1), we conduct experiments that involve changing the number of steps taken for optimization of a single prediction. Second, corresponding to the ability to verify predictions (Facet 2), we generate N predictions from an EBM and choose the minimum energy prediction (BoN or Self-Verification, which is formalized in Algorithm 2). This is conceptually similar to Best of N (BoN) sampling using language models Stiennon et al. (2020). However, EBMs generalize this approach to both discrete and continuous modalities and perform it on *every single prediction*, not just to entire sequences. We demonstrate performance improvements gained from both of these techniques in several Thinking experiments (Figures 7, 6, and B.6), which further confirm the importance of the described cognitive facets. This thinking process is formalized in Algorithm 2 and we more formally define and justify usage of the term thinking in Section C.1.

### 3.4 ENERGY-BASED TRANSFORMERS (EBTS) ARCHITECTURE

Transformers excel across domains due to their parallelizability, stability, and scalability Borsos et al. (2023); Oquab et al. (2023); Radford et al. (2019); Vaswani et al. (2017). In contrast, Energy-Based Models (EBMs) struggle with these aspects Du & Mordatch (2019); Du et al. (2020); Li et al. (2023), making Transformers a natural fit for scaling EBMs. Consequently, we introduce Energy-Based Transformers (EBTs), Transformer implementations designed for EBMs. We developed two variants: a GPT-style Radford et al. (2018) causal decoder-only EBT, for autoregressive modeling, and a bidirectional EBT with full sequence attention Devlin et al. (2019); more details are in Section C.6. While the bidirectional EBT implementation is straightforward, the autoregressive EBT requires care to prevent information leakage (see Section C.6).

## 4 EXPERIMENTATION AND RESULTS

We experiment with EBTs across both Autoregressive (AR) Radford et al. (2019) as well as bidirectional models Devlin et al. (2019) in discrete and continuous spaces.[4] In discrete spaces, we focus on the language modeling objective. In continuous spaces, we focus on vision tasks of next frame prediction (Section B.1) and image denoising. All models are pretrained from scratch, as EBT's architecture is incompatible with existing foundation models, and therefore cannot be fine-tuned. We focus

---

[4]Here, autoregressive and bidirectional refer to the procedure for generation. It's worth noting that autoregressive models are compatible with bidirectional attention, as in Li et al. (2025b).

Table 2: **System 2 Thinking Ablations.** All *energy landscape regularization* techniques described in Section 3.3 and their impact on System 2 Thinking performance, measured by percent perplexity improvement. Thinking Longer denotes more optimization steps and Self-Verification denotes optimizing many predictions and choosing the best. Removing regularization, such as Langevin Dynamics, results in less energy landscape exploration, which improves single path performance (thinking longer) at the expense of self-verification performance.

| Model | Thinking Longer ↑ | Thinking Longer and Self-Verification ↑ |
|---|---|---|
| No Random Step Size | -1.47 | 0.19 |
| No Random Num. Steps | 0.00 | 9.65 |
| No Langevin Dynamics | **17.2** | 17.0 |
| No Replay Buffer | 14.8 | 17.8 |
| **Full System 2 Configuration** | 7.19 | **18.7** |

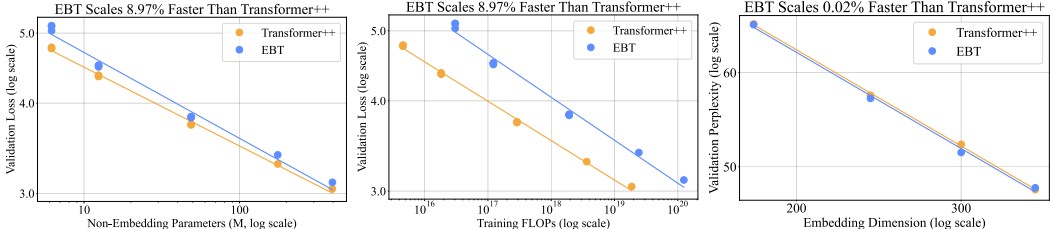

(a) Scaling for number of Parameters.  (b) Scaling for number of FLOPs.  (c) Scaling for the embed. dimension.

Figure 5: **Language Learning Scalability—Parameters, FLOPs, and Width.** Pretraining scaling comparisons between the Transformer++ recipe Touvron et al. (2023) and EBTs across model size (parameters), compute (FLOPs), and width (embedding dimension). EBTs have an 8.97% higher scaling rate than the Transformer++ in FLOP and parameter scaling (a and b), suggesting that EBTs offer promise as a pretraining approach.

on two primary types of results. First, we examine **learning scalability**, investigating how quickly models can fit the pretraining data, which is standard in pretraining Gu & Dao (2023); Hoffmann et al. (2022); Kaplan et al. (2020); Touvron et al. (2023). Second, we study **thinking scalability**, or how model performance changes as we scale the System 2 Thinking of models (Definition C.1), measured with the Number of Function Evaluations (NFEs) Chen et al. (2018); Ma et al. (2025) (forward passes).

### 4.1 Autoregressive Language Modeling Experiments

In this section, we detail and discuss the results for all NLP experiments using Autoregressive (AR) Language Models trained to predict the next discrete token in a text sequence Radford et al. (2019). All language models are pretrained on the RedPajamaV2 text corpus Computer (2023); Weber et al. (2024) $100B$ sample from HuggingFace using the GPT-NeoX tokenizer Black et al. (2022b) (as in Gu & Dao (2023)). Following existing pretraining work, we compare AR EBT with the standard Transformer++ recipe Gu & Dao (2023); Sun et al. (2024); Touvron et al. (2023).

For downstream evaluation, we used four key datasets in addition to the pretraining dataset, spanning reasoning, question answering, and syntax understanding. Ordered roughly by increasing perplexity difficulty, these include GSM8K Cobbe et al. (2021), SQuAD Rajpurkar et al. (2016), BigBench Elementary Math QA Srivastava et al. (2022), and BigBench Dyck Languages Srivastava et al. (2022). We focus on reasoning benchmarks due to their close alignment with System 2 Thinking.

We conduct scaling experiments for six different axes—including data, batch size, depth, parameters, FLOPs,[5] and embedding dimension. The results for the data, batch size, and depth scaling are shown in Figure 4; and the results for parameters, FLOPs, and embedding dimension are visualized in Figure 5. Across all axes, EBTs consistently have a higher scaling rate than the Transformer++ recipe, suggesting that EBTs offer promise at large scale[6].

---

[5]The FLOP calculation is nuanced and depends on specific hyperparameters, please refer to Section D.5.

[6]We cannot run larger-scale experiments due to limited computational resources, so larger-scale performance remains speculative.

Table 3: **Language Model Task Generalization Comparison.** We conduct experiments aimed at demonstrating the generalization of EBTs. Despite having slightly higher pretraining perplexity, EBTs often achieve lower perplexity on downstream tasks than the Transformer++, indicating better generalization. All models are trained with the same amount of data and parameters, but because EBTs at the current scale are less FLOP efficient (see Figure 5b), they used more FLOPs for this experiment. BB stands for BigBench.

| Model | Pretrain | GSM8K ↓ | SQuAD ↓ | BB Math QA ↓ | BB Dyck ↓ |
|---|---|---|---|---|---|
| Transformer++ | **31.36** | 49.6 | **52.3** | 79.8 | 131.5 |
| EBT | 33.43 | **43.3** | 53.1 | **72.6** | **125.3** |

Building on the learning results, we investigate EBTs for thinking at inference time. We found that the thinking capabilities of EBT emerge with a sufficiently large data scale, and therefore, due to limited resources, we focus on conducting thinking experiments with smaller models trained on substantial amounts of data. In Table 2 we conduct ablation studies to confirm the benefits of our *energy landscape regularization* techniques for System 2 Thinking on Out-of-Distribution Data from the BigBench Dyck Languages benchmark Srivastava et al. (2022). We find that using all techniques yields the best System 2 Thinking performance when combining extended thinking and self-verification. Additionally, the results show that randomizing the step size is critical—removing it nearly eliminates thinking gains. In contrast, disabling Langevin Dynamics degrades combined performance but improves results without verification, offering a performance-compute tradeoff.

Having established the importance of these landscape regularization techniques, in Figure 7, we analyze the scalability of thinking with EBTs, where the results yield two main insights. First, as shown in Figure 7a, EBTs are able to improve performance by as much as $29\%$ by increasing the amount of forward passes (thinking time), whereas the Transformer++ cannot improve performance.[7] This aligns with our claims that because traditional feed-forward Transformers cannot dynamically allocate additional computation for each prediction being made, they are unable to improve performance for each token by thinking for longer.

Second, as demonstrated in Figure 7b, the thinking capabilities of EBTs scale, showing that as EBTs are trained for longer, their ability to achieve improvements from verification improves, increasing up to $12\% - 14\%$ from $4\% - 8\%$. This suggests that EBTs trained at the same scale as modern foundation models, such as the 15T tokens Llama3 Grattafiori et al. (2024) was trained on ($\approx 1000\times$ the current scale), could have more substantial self-verification results.

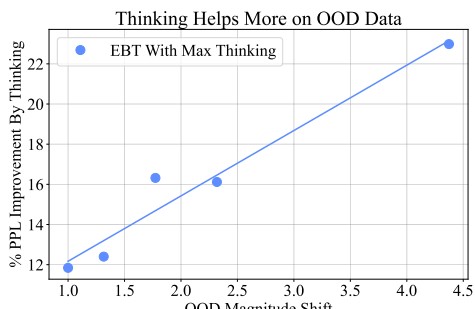

Figure 6: **OOD Thinking Performance.** As data becomes more OOD, thinking with EBTs leads to greater performance improvements—highlighting how thinking is critical for generalization to OOD data. Performance is measured on 5 datasets varying in Out-of-Distribution (OOD) magnitude shift, which is measured as the ratio of downstream perplexity to pretraining perplexity. Max Thinking combines thinking longer and self-verification.

As System 2 Thinking in humans is associated with generalization to novel scenarios, we conduct experiments directly aimed at measuring the effects of System 2 Thinking on generalization. In Figure 6, we visualize the performance of EBTs on the datasets described, which have varying levels of Out-of-Distribution (OOD) shift (measured as the ratio of downstream task perplexity to pretraining perplexity). We observe a strong linear trend: as the data becomes more OOD, thinking leads to greater performance improvements. Therefore, these findings suggest that the benefits of EBTs' thinking are not uniform across all data, but scale positively with the magnitude of distributional shifts, highlighting thinking as a critical mechanism for robust generalization. These findings align with observations in psychology, where humans rely on deliberate System 2 Thinking to tackle challenging OOD tasks.

Next, we investigate the relation between OOD generalization and pretraining performance. To investigate this, we compare models with identical training setups with respect to data and parameters,

---

[7]Because we pretrained language models from scratch, and are unable to train models the size of modern foundation models, we find models did not benefit from inference time techniques such as Chain-of-Thought.

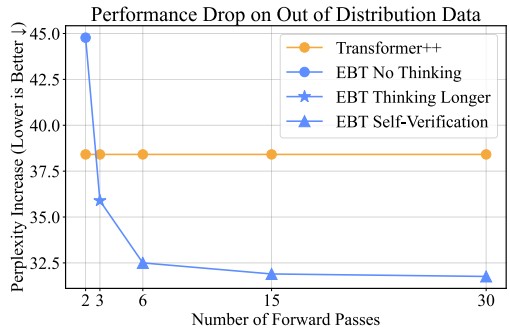
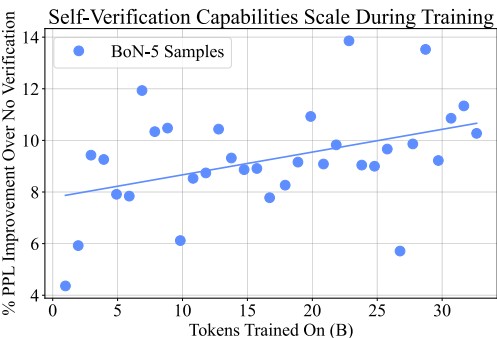

(a) OOD Thinking Performance Comparison.

(b) Verification Capabilities as Scale Increases.

Figure 7: **EBT Thinking Analysis.** (a) Mean performance degradation of the standard Transformer++ recipe Touvron et al. (2023) and the Energy-Based Transformer (EBT) on four Out-of-Distribution (OOD) datasets. The Transformer++, not being explicitly designed for inference-time computation, is unable to reduce perplexity at a *per-token level*. Alternatively, EBTs can improve performance with more forward passes over a single token/sample (Longer Thought) as well as generating many samples and choosing the minimum energy one (Self-Verification). (b) The performance of EBTs with and without self-verification; as scale increases, the benefits from self-verification increase. These results suggest EBTs generalize OOD better than the Transformer++ because of their System 2 capabilities, and that the thinking capabilities of EBTs improve during training.

Table 4: **Image Denoising and Classification Comparison**. For image denoising, EBTs significantly outperform DiTs Peebles & Xie (2023) in Peak Signal to Noise Ratio (PSNR), as well as MSE, on both in-distribution and Out-Of-Distribution (OOD) data, while using 99% less forward passes. On image classification, EBTs also perform better than DiTs, yielding around 10× higher accuracy, suggesting that EBTs learn better image representations and therefore understand images better than DiTs.

| Model | In Distribution Noise | | OOD Noise | | Image Classification | |
| | PSNR ↑ | MSE Pixel ↓ | PSNR ↑ | MSE Pixel ↓ | Top 1 Acc. ↑ | Top 5 Acc. ↑ |
| --- | --- | --- | --- | --- | --- | --- |
| DiT | 26.58 | 142.98 | 19.56 | 718.7 | 0.31% | 1.36% |
| EBT | **27.25** | **122.55** | **23.29** | **305.2** | **5.32%** | **13.2%** |

where EBTs have slightly worse pretraining perplexity than Transformer++ models. As shown in Table 3, despite achieving a higher pretraining perplexity, EBTs achieve lower (better) perplexity on most downstream tasks, suggesting stronger generalization, particularly to OOD data. Together, with the better learning scalability results, and knowing that improved pretraining performance usually leads to improved downstream task performance Chen et al. (2024); Isik et al. (2024), these results suggest that EBTs offer promise at scale during both pretraining and inference.

## 4.2 BIDIRECTIONAL IMAGE EXPERIMENTS

In addition to investigating autoregressive EBTs, we explore the performance of EBTs trained bidirectionally. Following Chen et al. (2020); Du et al. (2022), models are trained to denoise images with a fixed noise level. At inference, we test both the training noise level and a higher OOD level. The results are in Table 4, where we observe that EBTs perform better than DiTs at both in and out of distribution image denoising across various metrics. Following Chen et al. (2018), we plot the performance based on the number of forward passes (NFEs) in Figure B.6. These results demonstrate that EBTs perform better than DiTs while using 99% less denoising steps. Lastly, qualitative results for denoised out-of-distribution images for EBT compared to the DiT baseline are shown in Figure 8, demonstrating the improved visual quality of denoised images from EBTs.

In an effort to understand whether the representations learned from denoising captured useful visual features, we perform a linear probe evaluation on ImageNet-1k Russakovsky et al. (2015) of the models learned from denoising, following common practice in visual representation learning Oquab et al. (2023). The results are shown in Table 4, where the accuracy of EBTs is around 10× higher than that of DiTs, demonstrating that EBTs learn better image representations than DiTs.

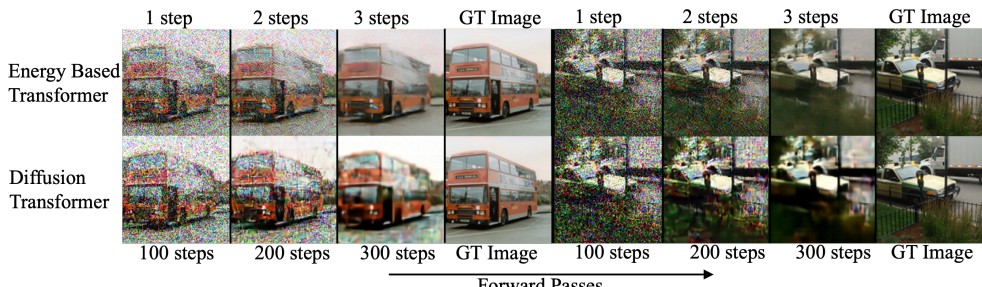

Figure 8: **Qualitative OOD Image Denoising.** EBTs achieve better denoising quality during inference while using one step for every 100 denoising steps of a DiT, or 99% less steps. The overall image quality of EBT denoised images is less blurry than images denoised by DiT.

## 4.3 DATA-CONSTRAINED EXPERIMENTS

To determine how EBTs perform in data-constrained settings, we conduct experiments aimed at measuring generalization with a small training set that is easy to overfit. We follow existing work benchmarking models on Sudoku Du et al. (2024), where we use the dataset from SAT-Net Wang et al. (2019) for the training set and the dataset from RRN Palm et al. (2018) as the test set. Models are trained to, given an initial Sudoku board with discrete numbers in the range of $1 - 9$ filled in, predict the rest of the Sudoku board that satisfies the necessary constraints. We report the accuracy on the test-set in Table 5, where we observe EBTs score the highest at 29.7%, RNNs score the second highest at 17.7%, and Feed-Forward Transformers score the worst at 0.03%. More details on these experiments are discussed in Section D.

Table 5: **Sudoku OOD Test Set Performance.** We compare a Feed-Forward (FF.) Transformer, an RNN based on recent work Jolicoeur-Martineau (2025), and an EBT on data-constrained algorithmic reasoning using a Sudoku dataset Palm et al. (2018).

| Architecture | Accuracy |
|---|---|
| FF. Transformer | 0.03% |
| RNN | 17.7% |
| EBTs | **29.7%** |

For more experiments on video, uncertainty estimation, and other topics, please refer to Section B.

## 5 LIMITATIONS AND CONCLUSION

**Limitations.** Despite demonstrating strong preliminary results, EBTs have several limitations. First, because EBTs generate predictions through an optimization process, they introduce additional hyperparameters. Second, while EBTs scale well up to 800M parameters, larger models were unexplored due to resource constraints. Third, for distributions that are highly multimodal, such as images, EBTs with the current formulation struggle to capture the many modes, hence why we often combine EBTs with autoregression. Finally, current EBTs lag behind feed-forward Transformers by a large margin in FLOP-efficiency, posing a high barrier to short-term adoption. Future researchers interested in leveraging EBTs will need to weigh the tradeoff between using more computation and improved generalization and reasoning.

**Conclusion.** We introduced Energy-Based Transformers (EBTs), a new approach that frames System 2 Thinking as an optimization procedure with respect to a learned verifier (an Energy-Based Model), enabling System 2 Thinking to emerge across many problems and modalities from unsupervised learning. Across discrete and continuous modalities, our results demonstrate that EBTs scale at a faster rate than the Transformer++ during pretraining across all measured axes, including data, batch size, depth, parameters, FLOPs, and width—with an up to 35% higher scaling rate. This suggests that EBTs offer promise at larger scale, even without System 2 Thinking. With System 2 Thinking, EBTs improve even further—increasing performance by up to 29% on text tasks, which we observe increases with data that is more Out-of-Distribution (OOD). Comparisons to DiTs on image denoising also reveal significantly better thinking scalability: EBTs match or exceed DiT's performance with only 1% of the forward passes. EBTs also learn substantially better representations, achieving approximately $10\times$ higher accuracy than DiTs. Ultimately, the improved scaling of EBTs during both training and inference positions them as a promising new approach.

## 6 ACKNOWLEDGEMENTS

We extend special thanks to Jeonghwan Kim and Cheng Qian for their helpful discussions. This work is based upon work supported by the U.S. National Science Foundation Graduate Research Fellowship Program under Grant No. DGE 21-46756, U.S. DARPA ECOLE Program No. #HR00112390060, DARPA ITM Program No. FA8650-23-C-7316, NSF Molecule Maker Lab Institute, an AI Institute for Molecular Discovery, Synthesis Strategy, and Manufacturing funded by the U.S. National Science Foundation under Awards No. 2019897 and 2505932, the AI Research Institutes program by National Science Foundation and the Institute of Education Sciences, U.S. Department of Education through Award No. 2229873 - AI Institute for Transforming Education for Children with Speech and Language Processing Challenges, and NSF NAIRR award. Any opinions, findings and conclusions or recommendations expressed in this material are those of the author(s) and do not necessarily reflect the views of the U.S. Government or the National Science Foundation. The U.S. Government is authorized to reproduce and distribute reprints for governmental purposes notwithstanding any copyright annotation therein. This research used the Delta and DeltaAI advanced computing and data resources, which are supported by the National Science Foundation (award OAC 2320345 and award OAC 2005572) and the State of Illinois. Delta and DeltaAI are joint efforts of the University of Illinois Urbana-Champaign and its National Center for Supercomputing Applications.

**Reproducibility Statement** We include the source code for full reproducibility of our results, which contains a detailed README on how to run and execute the code. We also include comprehensive experimental details for results in all Figures and Tables in Sections 4 and D. This includes all necessary hyperparameters, seeding, and model configurations for full reproducibility.

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

In this appendix, we provide additional insight and details on EBTs. First, we provide more insight on the broader impact/future work of EBTs in Section A. Next, in Section B, we include additional experiments. Then, we include additional approach details in Section C, as well as additional experimental details in Section D. After that, we include a comprehensive related work in Section E, additional facets of cognition in Section F, and a discussion of counterarguments in Section G. Finally, in the hopes of making EBTs more accessible to general audiences, in Section H we contain a general easier-to-understand intro to EBMs, and in Section I we describe a tutorial for getting started with EBTs.

**Large Language Model Usage Statement:** We use Large Language Models (LLMs) to assist with grammar, formatting, and writing clarity/flow in the paper. All intellectual and experimental contributions were made by the authors.

## A FUTURE WORKS AND BROADER IMPACT

### A.1 REVERSAL CURSE

Recently, a phenomenon known as "The Reversal Curse" has been observed where LLMs fail to learn a symmetric mapping Berglund et al. (2023) "B is A" despite learning "A is B". For example, LLMs trained on an example such as "Q: Who is Tom Cruise's mother? A: Mary Lee Pfeiffer" often fail to generalize to know the answer to the reverse question of "Who is Mary Lee Pfeiffer's son?" Remarkably, the Reversal Curse has manifested itself in LLMs regardless of the size or scale Berglund et al. (2023)—prompting researchers to investigate whether there are fundamental limitations to traditional feed-forward LLMs. One predicted cause of The Reversal Curse is the nature of gradient updates, where backpropagation only updates tokens within context. That is, while learning the mapping "A is B", none of B's tokens are within context, meaning they do not receive gradient updates when updating A's tokens. We hypothesize that LLMs trained with EBTs rather than standard feed-forward Transformers could help reduce this phenomenon, as with EBTs the tokens of A and B are within context during gradient updates due to predictions being made in the input space. Therefore, an exciting research direction would be investigating whether this hypothesis is correct in LLMs trained with EBT, allowing improved generalization.

### A.2 IMPROVED STABLITY

In this work, we primarily trained EBT with either two or three optimization steps. While these parameters worked well, we suspect that increasing the number of steps would improve the System 2 Thinking capabilities and the scaling during pretraining of EBTs, as more steps enable a longer "thinking process" before needing to converge. However, because of challenges in stability when training with more steps, due to a larger gradient computation graph, we were unable to successfully increase past two or three steps. Future work could focus more on extending the number of steps by studying ways to improve the training stability of EBMs.

### A.3 WORLD MODELS

In this work, we focus on autoregressive and bidirectional models over just state information (no actions). EBTs offer high promise in modeling states and actions due to the nature of EBMs learning a distribution over all possible inputs. Particularly, given a model trained to estimate the unnormalized joint distribution of the current context, future, as well as future actions, such world models could implicitly be used as policies to generate actions to achieve a specific state, similar to Chi et al. (2023); Janner et al. (2022); Zhou et al. (2024). This would involve holding the current context (past states) constant, and minimizing the energy by propagating the gradient back to the action inputs and future state predictions. Thus, world models trained in this manner become capable of more than just predicting the future, but also in decision making to achieve a specific goal state.

### A.4 EBTS AS COMPLEMENTARY MODELS

As demonstrated in Bakhtin et al. (2021); Bhattacharyya et al. (2020), EBMs can be used to improve the quality of generated text from language models. It's possible EBTs could be used in a similar

manner for a broad variety of tasks, serving as the verifier of predictions initialized by standard feed-forward models. Therefore, although we do a side-by-side comparison to existing models in this work, EBTs could be **complementary** to existing modeling approaches—being used as the System 2 Backbone for helping lighter models that perform System 1 thinking.

There exist several current real-world use cases, such as low-latency LLM serving, where doing a single forward pass is sufficient, and where the added inference overhead of gradients with EBTs would not be worth the extra computation. However, we also envision a world in which people use EBTs for long-term System 2 Thinking to solve challenging problems. How much computation would it be worth dedicating to prove a long-standing mathematical conjecture, or figuring out a cure to cancer?

### A.5 RECURRENT ENERGY-BASED MODELS

While EBTs scale well, for latency-driven use cases, Transformers require significantly more memory than Recurrent Neural Networks. Additionally, there is strong evidence for recurrence in the human brain Douglas & Martin (2007). Therefore, we anticipate that recurrent Energy-Based Models, possibly leveraging the Mamba architecture Gu & Dao (2023) will eventually become common.

### A.6 IMPROVED THINKING ALGORITHMS

The EBM thinking algorithms described have strong connections to or are derived from Markov Chain Monte Carlo (MCMC) sampling. Therefore, we broadly expect known MCMC samplers with more advanced techniques for traversing the energy landscape to be successful, such as Hamiltonian Monte Carlo Betancourt (2017) or annealed Langevin dynamics Du & Mordatch (2019). Additionally, we did not explore more advanced search algorithms such as Monte Carlo Tree Search, which we suspect could offer performance improvements and leave for future work.

### A.7 MULTIMODAL ENERGY-BASED MODELS

We did not experiment with multimodal EBMs, however, EBMs offer several advantages for learning over multiple modalities. For example, multimodal EBMs would enable a single energy scalar to represent the alignment between modalities, and would simplify joint training across modalities by providing a unified objective that naturally captures inter-modal dependencies.

### A.8 THINKING SCALABILITY

Due to a lack of computational resources, we were unable to train models with more than $10^{21}$ FLOPs ($\approx$ 1300 A100 GPU Hours). Therefore, training and thinking with EBTs remains untested at larger foundation model scale. We leave it to future work to scale with more GPUs and investigate the qualitative differences in training and thinking with EBTs.

### A.9 LEARNING MULTIMODAL DISTRIBUTIONS

We found that EBTs, with the current training approach, struggle to capture distributions with many modes (e.g., unconditional image generation or unconditional video generation). We believe this has to do with the current optimization-based formulation used during training. Therefore, future work could explore approaches to improve the learning of distributions with many modes, via changing the training formulation to support more modes. More info is in Section B.3.

### A.10 LEARNING BIDIRECTIONAL EBTS OVER ENTIRE SEQUENCES

Because of the described challenges with training EBTs for very multimodal distributions requiring uncertainty, we often trained EBTs combined with autoregression. Future work could focus on bidirectional EBTs operating over entire sequences, resulting in verification being more meaningful.

## A.11 UNDERSTANDING PREDICTIONS

We posit that there exists a fundamental distinction between the internal representations associated with model inputs and outputs. Specifically, models generate internal representations **of** inputs, as these serve as the foundation that dictate the model's behavior at any given point in time. Conversely, as outputs do not affect a model's behavior at a given point in time, we contend that models construct representations **for predicting** (and not necessarily understanding) outputs. This distinction leads us to a consequential insight: *models may not achieve a genuine understanding of outputs in the same way they understand inputs.* This implies that while models may develop an intricate understanding of input data, such understanding does not naturally extend to predictions that are made in the output space. Therefore, existing feed-forward models primarily making predictions in the output space may not **understand** their predictions in the same way they understand their inputs. This intuition further supports the principles behind EBT, where predictions are made in the input space enabling representations **of predictions** to be developed. We leave the investigation of this hypothesis to future work.

## A.12 SOCIETAL IMPACT

The ability to achieve human-like thinking with AI offers benefits in multiple domains. As such, EBTs offer several potential positive impacts, through the enabling of AI to potentially think more like humans. On the other hand, it's also possible that more intelligent AI models trained using EBT could be misused for harm by malicious actors.

# B ADDITIONAL EXPERIMENTATION

## B.1 AUTOREGRESSIVE VIDEO EXPERIMENTS

To assess how EBTs scale in continuous domains, we train models to predict the next image in a video conditioned on all previous frames—a common pretraining objective for video models Deng et al. (2024b); Gu et al. (2025); Rakhimov et al. (2020); Weissenborn et al. (2019); Ye & Bilodeau (2023). Unlike in the NLP experiments, where models see each sample only once due to the dataset size, current popular video datasets are relatively small, requiring models to train repeatedly on the same data. As a result, this setting probes a different question: *"how well can models fit a fixed dataset?"*, rather than how efficiently models scale under non data-bound regimes. This distinction is especially important given the recent scarcity of high-quality datasets Dat; Villalobos et al. (2022) and the perspective that data will increasingly become a bottleneck.

For experiments, we encode all $224 \times 224$ images into 3136 dimensional features with the frozen SD-XL VAE Rombach et al. (2022); Stability. Then, all models are trained using a Smooth L1 loss with $\beta_{\mathcal{L}} = 1.0$ on the Something Something V2 dataset Goyal et al. (2017), where we report the minimum validation loss achieved. In Figure B.1 we report scaling results for the embedding dimension and non-embedding parameter count, as we found these axes behaved the most linearly. The results demonstrate that, despite achieving a higher initial loss, EBTs scale at a more than $33\%$ faster rate than the Transformer++. This suggests that at a larger scale, EBTs could achieve better performance than the Transformer++.

We believe this large scaling rate gap can be linked to the fact that EBTs more seamlessly model continuous distributions than standard feed forward transformers due to being able to express uncertainty (Facet 3) through their energy scalar. To confirm this, we visualize results for different energies when predicting video frames in Figure B.2. The results demonstrate that EBTs successfully learn to capture uncertainty—where frames earlier on in the video have higher energy (higher uncertainty) due to no large objects being within the frame, and then as the major object in the scene becomes revealed more EBT predicts lower energy (lower uncertainty). EBTs learn to exhibit this behavior without any supervision using a Smooth L1 loss, whereas the standard feed-forward Transformer++ would require discretization schemes such as Vector Quantization Islam et al. (2023) with a categorical loss, or other tricks to achieve the same effect.

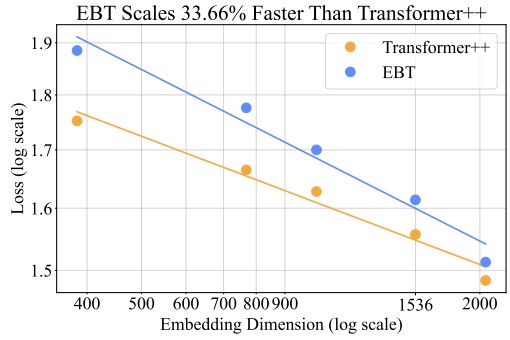

(a) Scaling for the embedding dimension.

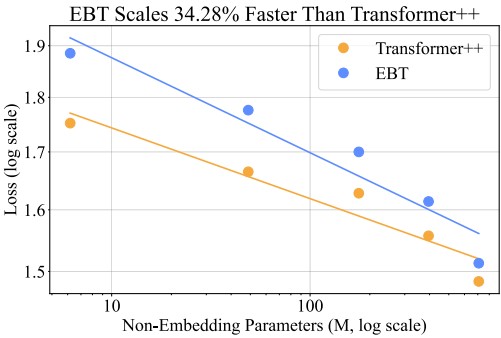

(b) Scaling for the Non-Embedding Parameters.

Figure B.1: **Video Learning Scalability—Width and Parameters.** The minimum validation loss achieved on the Something Something V2 (SSV2) dataset. While EBTs achieve higher validation loss than the Transformer++ at smaller scales, the scaling rate is more than $33\%$ higher, suggesting that at larger scales EBTs could perform better than the Transformer++. Notably, scaling with respect to the embedding dimension behaves more linearly than for the number of parameters, likely due to the embedding dimension serving as a bottleneck for the image representation.

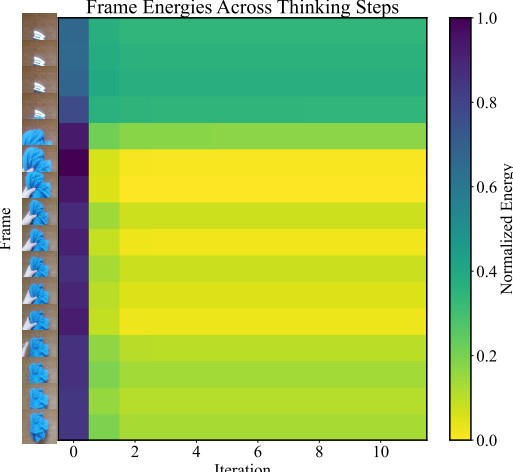

Figure B.2: **Learning Uncertainty on Video Results.** In line with cognitive Facet 3, EBTs learn to express uncertainty across continuous video frames without supervision. At the start of the video, uncertainty is high (high energy) because the frame is mostly empty and the scene is highly unpredictable. As a blue garment is placed into the frame, uncertainty decreases (low energy), reflecting the greater predictability of the scene. When the blue garment is removed from the scene, uncertainty increases again, indicating a return to higher unpredictability. Such a capability is significantly more difficult to achieve in continuous spaces with traditional feed-forward transformers without discretization schemes Pei et al. (2022).

## B.2 Additional Natural Language Processing Experiments

We conduct experiments to confirm hypotheses on thinking results obtained in the main paper. First, we confirm that EBTs become less adversarial with scale by comparing the performance of using Best-of-N (BoN) with 2 versus 10 samples. The results in Figure B.4a demonstrate that when models are trained on less tokens, there is little performance improvement by verifying 10 samples instead of just 2. In fact, verifying 10 samples occasionally leads to *worse* performance than verifying 2 samples, likely because the EBT found an adversarial sample (a sample with low energy that is in fact not a good prediction). However, as data scale increases we observe that performance improvements from BoN-10 versus BoN-2 increase, and that these adversarial dynamics decrease. Together, these results suggest that with scale EBTs become less adversarial due to an improved energy landscape. In an effort to understand the impacts of thinking at the scale of modern foundation models, we

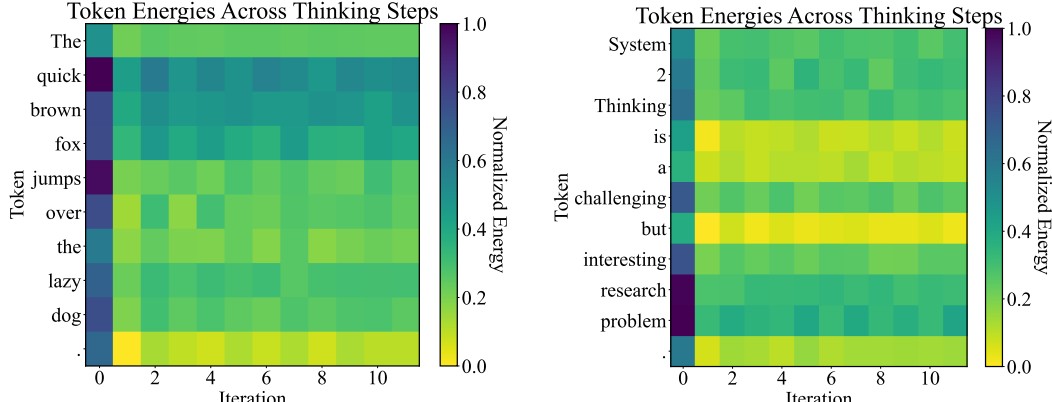

Figure B.3: **Learning Uncertainty on Text Results.** EBTs learn to vary uncertainty across text tokens without any explicit supervision. As an example, in both (a) and (b), simple tokens such as ".", "is", "a", "but", or "the" have lower energies across inference-time optimization (thinking) steps, indicating lower uncertainty. On the other hand, harder to predict tokens such as "quick", "brown", "research", and "problem" have higher energies across optimization steps, and more difficulty in achieving energy convergence, meaning the model is more uncertain. Inspired by Geiping et al. (2025).

project results from Figure 7b to the data scale of modern foundation models Grattafiori et al. (2024) to extrapolate a projected performance gain from self-verification based thinking. The results are visualized in Figure B.4b, where, while being highly speculative, they demonstrate that, because of the $1000\times$ data scale of modern foundation models, the performance improvement from self-verification increases. While these results remain largely speculative, they demonstrate the potential for System 2 Thinking with EBTs that improves with scale.

Additionally, we visualize results from EBT at representing uncertainty while predicting tokens in Figure B.3. The results demonstrate that for easier to predict tokens, such as "the" or "but", EBTs optimize to lower energies faster, whereas for harder to predict tokens, such as "fox" or "problem" EBTs have higher energy that does not converge across steps. This suggests that during pretraining EBTs learn to capture uncertainty regarding which tokens are harder or easier to predict, achieving Facet 3. In addition to learning aleatoric uncertainty (uncertainty related to noise), in an effort to understand whether EBTs can capture epistemic uncertainty (uncertainty related to knowledge), we visualize the energies of different token sequences that are in versus Out-Of-Distribution (OOD) in Figure B.5. The results demonstrate that, for a more in-distribution sequence, EBTs have lower energy (less uncertainty), than for an OOD sequence. This suggests that EBTs learn to *know what they don't know*, as they learn to have higher energy for OOD sequences signifying harder predictability. This is a promising characteristic of EBTs, as it enables uncertainty estimation within continuous state spaces, enabling principled inference-time behavior adaptation (Facet 1) when models sense a more challenging problem.

### B.3 EBT FAILURE CASES

During experimentation, along with image denoising experiments, we also conducted small-scale experiments with text-to-image generation. We found that, in datasets such as COCO with many different modes for a single condition (e.g., hundreds of images with a caption similar to "giraffe with a long neck"), that EBTs did not learn to generate high-quality novel images. Instead, EBTs often generated blurred images similar to the training distribution. We believe this is caused by the training approach pushing the energy landscape to have a single local minimum surrounding the training examples. Therefore, when there are many different modes, these local minima within the energy landscapes "merge" to one minima averaged between the different modes, resulting in blurriness. We believe that this is not a fundamental limitation of EBTs, and that future work could address this issue.

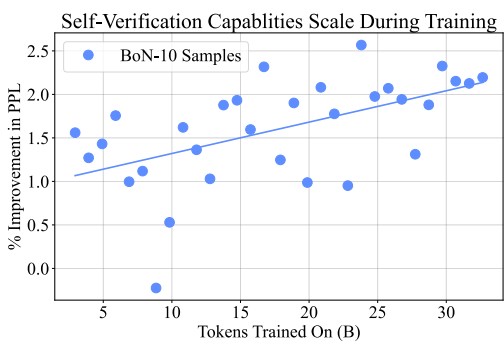

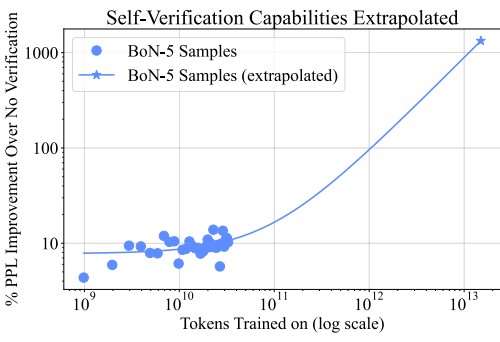

(a) Self-verification with BoN-10 versus BoN-2.

(b) Results in Fig. 7b projected to Llama3 scale Grattafiori et al. (2024).

Figure B.4: **EBT Thinking Analysis for Data Scaling.** (a) Self-verification of BoN-2 compared to BoN-10. EBTs become less adversarial and thus benefit more from verifying an increasing number of samples during training. (b) A projection of the results from Figure 7b to the data scale of Llama3 Grattafiori et al. (2024), demonstrating that as data scale increases, improvements from self-verification can lead to potentially massive performance increases from System 2 Thinking.

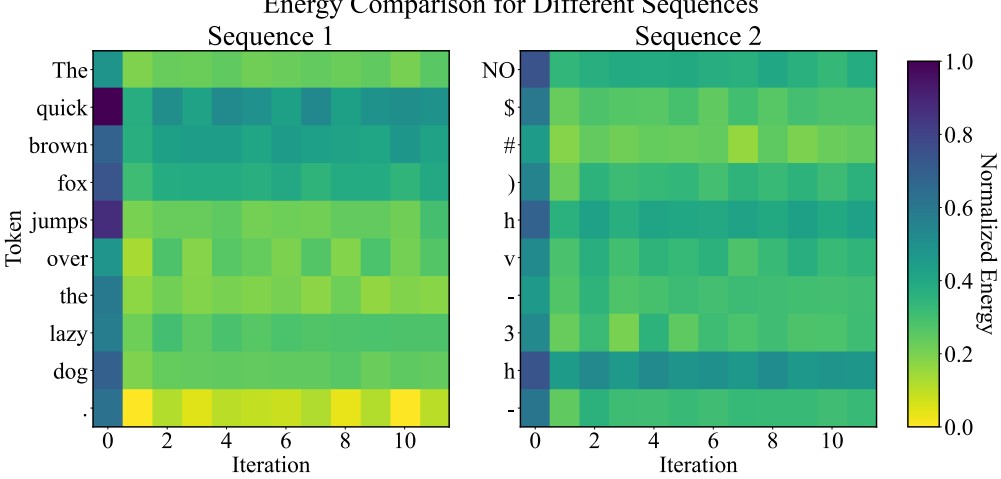

Figure B.5: **Epistemic Uncertainty Comparison.** EBTs learn to express epistemic uncertainty (uncertainty related to lack of knowledge) on unseen data. Particularly, the sequence on the left, which is a text sequence likely seen during training, has consistently lower energy (uncertainty) for tokens than the sequence on the right, which is a random text sequence not from the training distribution. This demonstrates that EBTs learn to "know what they don't know."

# C  ADDITIONAL EBT DETAILS

## C.1  FORMALIZING THINKING

Due to the recent surge of interest in scaling the performance of models during inference/test time, there are several common terms used to refer to these ideas. These include scaling the thinking capabilities of models Jaech et al. (2024), inference time scaling Ma et al. (2025), inference time compute Manvi et al. (2024), and test time compute Jaech et al. (2024); Snell et al. (2024). Therefore, to reduce confusion stemming from a wide variety of terminology and unite the community, in this work we broadly define these concepts as **System Two Thinking** or more simply **Thinking**. We formalize improvements made by Thinking as the following:

**Definition C.1** (System 2 Thinking). *Given a problem with data $x$, a model $\theta$, and additional computational resources in the form of function evaluations $F$ greater than the minimum number*

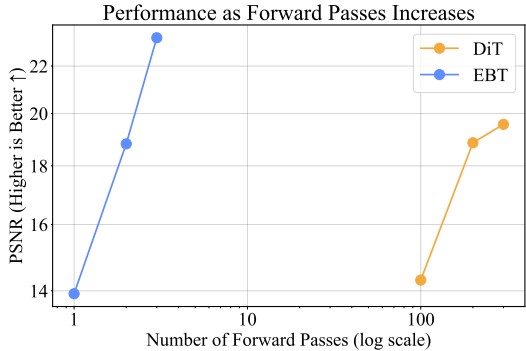

Figure B.6: **Image Denoising Thinking Scalability.** A comparison between EBT and DiT on image denoising given a different number of forward passes. EBTs require only $1\%$ of the forward passes used by DiT to achieve comparable or better PSNR. Further, the scaling rate of PSNR improvement given more forward passes is much higher for EBTs than it is for DiTs. These results suggest EBTs have superior thinking capabilities than DiTs on OOD data.

*of function evaluations to get a valid prediction from the model $F_0$, **System Two Thinking** $STT(\cdot)$ quantifies the expected percentage improvement in performance as $F$ increases. Let $P(x, \theta, F)$ be the performance on input $x$ when the model $\theta$ uses $F$ function evaluations:*

$$\text{STT}(x, \theta, F) \;=\; \mathbb{E}_x \left[ \frac{P(x, \theta, F)}{P(x, \theta, F_0)} \;-\; 1 \right],$$

This formalization is compatible with any type of metric (e.g., Accuracy, Perplexity, FID, etc), and uses more psychology-aligned terminology Kahneman (2011). Further, avoiding terms such as "inference" or "test-time" makes the idea of Thinking more compatible with domains where the line between inference and training is blurry, such as real-world continual learning, domain adaptation, or actual human learning/thinking processes Parr et al. (2022). Just as **learning** has become a flexible term across machine learning representing several different ideas, we intend for **thinking** to similarly unify many diverse ideas under a common framework. For a greater justification on this perspective, please see Section G

## C.2 ENERGY-BASED TRANSFORMER (EBT) THINKING TYPES

All experiments in the paper are conducted with two main variants of EBTs, which we call System 1 (S1) and System 2 (S2) EBTs. S1-EBTs have hyperparameters specifically optimized for stability and learning convergence, whereas S2-EBTs have hyperparameters optimized for System 2 Thinking capabilities. Many of the pretraining scaling experiments conducted in Section 4 are with S1 models as to reduce the computational resources required for experimentation. To confirm that these results hold for S2 models, we plot the scaling trends of S1 and S2 models side by side; in Figure C.1, we find that S2 models scale *at the same or a higher rate* than S1 models during training, but have a higher Y-intercept. This Y-intercept offset does not affect asymptotic scaling behavior (as asymptotically the scaling rate dominates), and hence, scaling trends that hold for S1 models hold for S2 models (S2 models may even perform better than S1 models during pretraining asymptotically because of the higher scaling rate). Intuitively, switching from S1 to S2 models allows for a compute trade-off between the model's pretraining performance and the model's System 2 Thinking capabilities.

S1 models have the gradient of predictions detached between optimization steps to increase training stability. Conversely, following Du et al. (2024), the S2 models truncate backpropagation and avoid detaching prediction tensors between optimization steps. Additionally, the S2 models have all the energy landscape regularization techniques described in Section 3.3, whereas the S1 models have none. We also find that S2 models require a different value for the optimization step size, that the optimization step size not be learned, and to perform a minimum number of steps (and not just any value from 1 to the max number of steps).

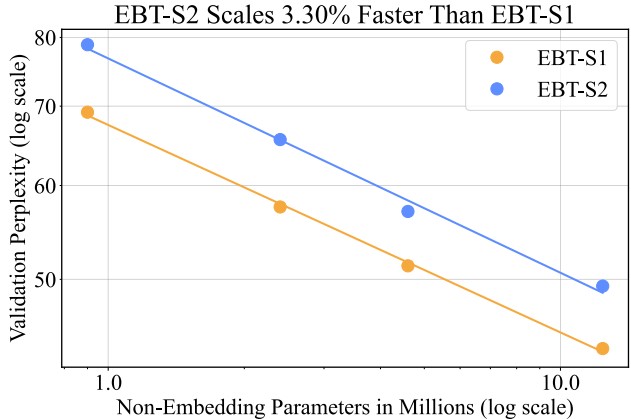

Figure C.1: **EBT S1 and S2 Scaling Comparison.** Scaling rate of System 1 (S1) compared to System 2 (S2) models. System 2 models have a higher Y-intercept but scale slightly faster than System 1 models. Therefore, as the scaling rate ultimately dominates asymptotic scaling behavior, and not the Y-intercept, scaling results in the main paper that hold for S1 models generally hold for S2 models.

### C.3 Energy-Based Transformer Toy Example

Similar to standard neural network training, the optimization process for EBTs starts from an initial random guess. For neural networks, this corresponds to randomly initialized weights; in our EBT formulation, it corresponds to initializing the prediction $y_0$ by sampling from a Gaussian distribution. In neural networks, we compute a loss and differentiate it to obtain a gradient with respect to the network parameters, which we then update to reduce the loss.

In contrast, our method optimizes the prediction directly. Given a current prediction $y_t$, we input $y_t$ (and the conditioning $x$, if applicable) into the EBM to obtain a scalar energy $E_\theta(x, y_t)$. We then compute the gradient of this energy with respect to the prediction, $\nabla_y E_\theta(x, y_t)$, and update the prediction via gradient descent to decrease the energy. This iterative procedure for updating the prediction mirrors the way neural network parameters are updated during training to minimize the loss, but operates in prediction space instead of parameter space.

### C.4 Avoiding the Curse of Dimensionality

This approach avoids the curse of dimensionality inherent in traditional contrastive Energy-Based Model (EBM) training by reframing learning as an optimization problem (Algorithm 1). Contrastive methods must explicitly push up the energy of an exponentially increasing number of negative samples in high-dimensional space. Our optimization-based approach, instead, implicitly shapes the energy landscape by training the network to successfully perform gradient descent from a random initial prediction to the ground truth solution. This implicitly regularizes the landscape to only have local energy minima at data points. Because regularized methods are known to suffer less from the curse of dimensionality, this successfully avoids such a pitfall LeCun (2022). This aligns with more formal results on Ising models from theoretical computer science, where learning the energy function (Hamiltonian) is of much simpler complexity-wise than accurately sampling or learn the full resulting probability distribution Ravikumar et al. (2010).

### C.5 Computing Second-Order Gradients

Because training EBTs involves supervising the gradient of the energy function, second-order gradients must be used. Using PyTorch Paszke et al. (2019), these second-order gradients are computed using `torch.autograd.grad` with `create_graph` set to `True`. This function keeps the graph from backpropagation in memory, such that the loss can be computed through this derivative.

Because this involves a derivative of a derivative, it requires second-order derivatives. Unfortunately, formulating the entire Hessian naively is expensive. Thus, deep learning frameworks such as PyTorch often automatically resort to computing Hessian Vector Products (HVPs) when possible. These HVPs are linear in computational time and memory, enabling scalable second-order computation whose cost is comparable to a normal backward pass Dagréou et al. (2024); Pearlmutter (1994).

### C.6 AUTOREGRESSIVE CAUSAL ENERGY-BASED TRANSFORMERS IMPLEMENTATION

In this section, we detail the implementation of decoder-only autoregressive EBTs, and the complexity that arises due to the way EBMs work. Particularly, the implementation of causal-attention decoder-only autoregressive transformers poses a challenge due to making predictions in the output space rather than in the input space. To demonstrate why this poses a challenge, for the decoder-only autoregressive transformer consider the case of the $n \times n$ attention scores matrix after the causal mask has been applied:

$$\text{scores} = \begin{bmatrix} \alpha_{z_1,z_1} & 0 & \dots & 0 \\ \alpha_{z_2,z_1} & \alpha_{z_2,z_2} & \dots & 0 \\ \dots & \dots & \dots & \dots \\ \alpha_{z_n,z_1} & \alpha_{z_n,z_2} & \dots & \alpha_{z_n,z_n} \end{bmatrix},$$

where $\alpha_{z_i,z_j}$ represents the attention score (probability mass) from state $z_i$ to state $z_j$. Now, in the case of an EBM, where predictions of future states are made in the input space, the intended $n \times n+1$ attention scores matrix would look like the following:

$$\text{scores} = \begin{bmatrix} \alpha_{z_1,z_1} & \alpha_{z_1,\hat{z}_2} & 0 & \dots & 0 \\ \alpha_{z_2,z_1} & \alpha_{z_2,z_2} & \alpha_{z_2,\hat{z}_3} & \dots & 0 \\ \dots & \dots & \dots & \dots & \dots \\ \alpha_{z_n,z_1} & \alpha_{z_n,z_2} & \alpha_{z_3,z_3} & \dots & \alpha_{z_n,\hat{z}_{n+1}} \end{bmatrix}. \tag{3}$$

This is challenging to compute because each $\hat{z}_i$ along the superdiagonal is unique for its row. Consequently, this matrix cannot be computed with a matrix multiplication (softmax $\left( \frac{QK^T}{\sqrt{d_k}} \right)$) as in regular attention, as every value on the superdiagonal is a prediction and not a past state.

Additionally, in a traditional transformer, if the context length is $n$, the size of the passed in tensor will be $bs \times n \times d$ where $bs$ is the batch size and $d$ is the embedding dimension. However, since EBMs make predictions in the input space, the input tensor needs to be different to allow for inputting future predictions. Therefore, for a context length of $n$, we define the first $n-1$ elements as $z_o$, or the original sequence representations, and the final $n-1$ elements as $z_p$, or the predicted sequence representations.

The values for $z_{o1}^n$, or the given states, are computed the same as in the original transformer Vaswani et al. (2017), as the attention scores of the original states do not depend on the predicted states. This can be formalized as the following:

$$\text{Attention}(Q_o, K_o, V_o)_{z_{o1}^n} = \text{softmax}\left( \frac{Q_o K_o^T}{\sqrt{d_k}} \right) V_0, \tag{4}$$

Where $Q_o$, $K_o$, and $V_o$ are the Query, Key, and Value matrices of the past states $z_{o1}^n$. In every block of the transformer, the representations of all past states are updated in this manner, independent of the representations of the predicted future states.

For the computation of representations over future states, 3 matrices are also computed, but for the representations of predicted future states rather than past states. We call these $Q_p$, $K_p$, and $V_p$. First, we compute the self-attention scores of all future representations to all past representations:

$$\text{unnormalized\_scores\_p} = \frac{Q_p K_o^T}{\sqrt{d_k}}. \tag{5}$$

Note, however, that the self-attention scores of each predicted future state with itself is not calculated–due to the key matrix being from the original states. Therefore, the superdiagonal needs to be replaced with the self-attention scores of each predicted future state with itself to achieve the attention score matrix shown in Equation 3.

EBT involves two separate tensors–one for past states and one for predicted future states. We denote these as $z_1^n$ and $\hat{z}_1^n$ where $z$ are known past states and $\hat{z}$ are predicted future states. The intended attention scores matrix is the following:

$$\text{scores} = \begin{bmatrix} \alpha_{z_1,z_1} & \alpha_{z_1,\hat{z}_2} & 0 & \dots & 0 \\ \alpha_{z_2,z_1} & \alpha_{z_2,z_2} & \alpha_{z_2,\hat{z}_3} & \dots & 0 \\ \dots & \dots & \dots & \dots & \dots \\ \alpha_{z_n,z_1} & \alpha_{z_n,z_2} & \alpha_{z_3,z_3} & \dots & \alpha_{z_n,\hat{z}_{n+1}} \end{bmatrix}.$$

To compute Equation 3, we first need to append a column to the right side of the unnormalized_scores_p matrix, as the size of the matrix is currently $n - 1 \times n - 1$, but we need to have $n$ representations within context. After doing this, we first mask out the superdiagonal, to ensure that the probabilities in the score matrix only correspond to the values of the predicted future states with itself. This masking operation is done through elementwise multiplication of a matrix with 1's everywhere except the superdiagonal, which has 0's. Then, we compute the self-attention scores of each predicted future state with itself, using the following equation:

$$\text{z\_p\_self\_attention} = sum(Q_p * K_p), \tag{6}$$

where the $*$ indicates the Hadamard product and the sum is across the fourth, attention head, dimension. Using a superdiagonal mask again, we set the diagonal of the unnormalized_scores_p to these values. Now, after applying the softmax:

$$\text{scores\_p} = \text{softmax}\left(\text{unnormalized\_scores\_p}\right), \tag{7}$$

we have the intended scores matrix shown in Equation 3. However, one more barrier towards finally extracting all updated $z_{p1}^n$ representations is the fact that we cannot simply multiply this resulting scores matrix by the values matrix, as each element of the superdiagonal corresponds to a different predicted next future state. Thus, using similar techniques to before, we first clone and then extract the superdiagonal from this scores matrix using a diagonal mask.

After extracting the superdiagonal, we can multiply the resulting scores matrix by the $V_o$ matrix to get all of the representations summed together of each predicted future state with all past states. This is represented as the following matrix multiplication:

$$z_{p1}^n = \text{scores\_p} \cdot V_o. \tag{8}$$

As we also need to add the representation of each predicted future state weighted with its own attention score (what was extracted on the superdiagonal), we perform another Hadamard product of the $V_p$ matrix with the cloned superdiagonal to get these values, and then add these element wise to the $z_{p1}^n$ representations. Now, we have computed the intended representations involving the scores matrix shown in Equation 3. Thus, $z_1^n$ and $\hat{z}_1^n$ are updated using $z_{o1}^n$ and $z_{p1}^n$ respectively, by multiplying these tensors by the output weight matrix $W_o$.

### C.7 Autoregressive Energy-Based Transformers Simplified Implementation

A more simplified implementation involves the entire attention matrices and a generalized causal mask, as described in Deng et al. (2024a). However, because this implementation involves a matrix multiplication with 2 times the sequence length, this results in 4 times the number of FLOPs as normal attention, which is around double the number of FLOPs of our more efficient implementation.

## D Experimentation Details

Tables D.3 and D.4 specify general model information and hyperparameters. We utilized the Llama 2 transformer implementation Touvron et al. (2023) for the Transformer++ and used this implementation as the backbone upon which we built EBT. We seed all libraries using PyTorch Lightning Falcon (2019) for all experiments with a seed of 33. For the Diffusion Transformer, we use the implementation from Peebles & Xie (2023)—for the bidirectional EBT we build upon this implementation.

### D.1 Autoregressive Language Modeling Experimental Details

For all scaling experiments, we copy the popular Mamba paper pretraining settings (for the model configuration, not necessarily for data and learning rate configuration), shown in Table D.2. Because we are compute-limited, we also include two extra model sizes in this table, extra extra small (xxs) and extra small (xs).

We manually created a training and validation split of 66 million and 33 thousand samples for the red pajama dataset, respectively. Additionally, we focus on reporting perplexity as our relatively small models trained from scratch, when compared to current foundation models, do not achieve high accuracies on many of the benchmarks used. Furthermore, perplexity often functions as a more linear metric than accuracy Gu & Dao (2023); Schaeffer et al. (2023), enabling a more comparable analysis of downstream performance as we scale compute during inference.

#### D.1.1 Learning Scalability Experimental Details

Conducting thorough scaling experiments is very challenging—a recent survey on "scaling laws" Li et al. (2025a) showed just how fragile many of these "scaling laws" are to hyperparameters, data, etc. and how changing these parameters slightly can lead to different conclusions. Being bottlenecked by a limited set of computing resources further exacerbates this issue. Therefore, we sought out to conduct controlled experiments that revealed the most information possible regarding the scaling of EBTs compared to different models.

Most existing works studying scaling by changing several factors at the exact same time, including depth (number of transformer blocks) Kaplan et al. (2020), width (embedding dimension) Kaplan et al. (2020), possibly batch size Chen et al. (2024); Hu et al. (2023), and the amount of data Kaplan et al. (2020). Therefore, to be more comprehensive in determining when EBTs scale differently than the Transformer++, we decided to conduct normal scaling experiments over all of these factors at the exact same time (as is standard), *as well as ablating over just changing just one of these parameters at a time*. Notably, conducting experiments in this manner allows for controlling a single independent variable at a time (i.e., just changing the number of Transformer Blocks), which allows for stronger conclusions regarding what aspects of model scaling different models perform better over (i.e., EBTs scale better then the Transformer++ when increasing the number of Transformer Blocks). Scaling all factors at once does not allow for such insight, as there are many independent variables.

For our parameter and FLOP scaling law experiments in Figures 5 a and b, we follow the popular and reliable Chinchilla scaling laws Hoffmann et al. (2022), where we scale the number of parameters and the data models are trained on proportionally. We use a factor of $20\times$ the amount of tokens as parameters.[8] We do this for XXS, XS, Small, Medium, and Large models in Table D.2. All model sizes use the same batch size of 64, and context length of 256 tokens, where we scale the number of training steps to increase data as parameter scale increases. We use three random seeds ($33 - 35$) for the XXS, XS, and Small models in order to improve robustness and reliability of our scaling laws while also reducing variance.

In addition to plotting the results in Figures 5 a and b, in order to increase transparency and reproducibility, we also report raw losses in Table D.1. This allows for fitting the scaling law exponents, which allows for computing the scaling rate difference in the two models used (EBTs and the Transformer++). Following Chinchilla Hoffmann et al. (2022), we model the scaling behavior as a power law $L(C) = \beta C^{-\alpha} + E$, where $L$ is the loss, $C$ is the compute budget (FLOPs), and $E$ is the irreducible entropy of natural text. Assuming $E$ is constant across models, we approximate the scaling rate by fitting a linear regression in the log-log regime:

$$\log(L - E) \approx -\alpha \log(C) + \log(\beta)$$

Here, $\alpha$ represents the scaling exponent. By comparing the exponents for EBTs and Transformers, we can determine which architecture improves loss more efficiently per unit of compute. Based on the fit linear regression, we also compute an $R^2$ coefficient to determine the reliability of these scaling laws for FLOPs, where we get a value of $99.69\%$ for the Transformer++ and a value of $98.40\%$ for

---

[8]For these experiments we transitioned to using the FineWeb dataset Penedo et al. (2024), which is of high quality and has recently become popular.

Table D.1: **Pretraining Scaling Law Loss Across Model Sizes and Seeds.** Cross-entropy loss for the Transformer++ and EBT models at five model sizes (XXS to Large) and three random seeds (33, 34, 35). Medium and Large models were trained with a single seed due to increased compute requirements.

| Size | Model | Seed 33 | Seed 34 | Seed 35 | Non-Embedding Params. | FLOPs |
|------|-------|---------|---------|---------|------------------------|-------|
| XXS | Transformer++ | 4.7786 | 4.7638 | 4.7619 | 6.18 | 4.56E15 |
| XXS | EBT | 5.1210 | 5.1122 | 5.0392 | 6.18 | 3.04E16 |
| XS | Transformer++ | 4.3728 | 4.3594 | 4.3546 | 12.4 | 1.82E16 |
| XS | EBT | 4.5119 | 4.5118 | 4.4906 | 12.4 | 1.22E17 |
| Small | Transformer++ | 3.7333 | 3.7354 | 3.7395 | 48.8 | 2.87E17 |
| Small | EBT | 3.8262 | 3.8321 | 3.8189 | 48.8 | 1.92E18 |
| Medium | Transformer++ | 3.2950 | – | – | 176 | 3.63E18 |
| Medium | EBT | 3.3922 | – | – | 176 | 2.42E19 |
| Large | Transformer++ | 3.0443 | – | – | 395.5 | 1.87E19 |
| Large | EBT | 3.1079 | – | – | 395.5 | 1.24E20 |

Table D.2: **Model sizes and hyperparameters for scaling experiments.** For most model sizes we follow Gu & Dao (2023).

| Size | Non-Embedding Params | # layers | embed. dim | # heads |
|------|----------------------|----------|------------|---------|
| XXS | 6.18M | 6 | 384 | 6 |
| XS | 12.4M | 12 | 384 | 6 |
| Small | 48.8M | 12 | 768 | 12 |
| Medium | 176M | 24 | 1024 | 16 |
| Large | 396M | 24 | 1536 | 16 |
| XL | 708M | 24 | 2048 | 32 |

EBTs for FLOPs, both of which are within well-accepted ranges for Chinchilla scaling laws Inbar & Sernau (2024).[9]

### D.1.2 THINKING SCALABILITY EXPERIMENTAL DETAILS

We train xxs models with the same setup as above, with the exception that models are trained with a batch size of 128 for 1M training steps. Increasing the data scale enables us to better understand how thinking scales during pretraining. It's worth noting that since we are training small language models, they could not benefit from modern techniques such as Chain of Thought (CoT) in improving performance.

### D.2 AUTOREGRESSIVE VIDEO EXPERIMENTAL DETAILS

We use the same model parameter scaling, shown in Table D.2, as the NLP experiments. We also used a batch size of 256 for all models, as we found that it did not significantly affect the scaling performance due to models training for many epochs. We also processed videos with 0.25 seconds between frames. For the Transformer++ baseline, we use the same learning rates as the NLP experiments. For EBT, we found that it was necessary to use a lower learning rate by a factor of 3. We use the standard SSV2 train and validation split for experiments. Other hyperparameters are shown in Table D.3 and Table D.4.

### D.3 BIDIRECTIONAL IMAGE DENOISING EXPERIMENTAL DETAILS

We use the COCO 2014 dataset Lin et al. (2014); Tawfik with 128 by 128 images, its train/validation split, a patch size of 16, and the Diffusion Transformer implementation from Peebles & Xie (2023).

---

[9]Note that due to not including all significant figures when reporting the losses and FLOPs, the $R^2$ coefficient from this table may be slightly different from our reported results.

Table D.3: **Hyperparameters for Transformer++.**

| Hyperparameter | CV | NLP |
|---|---|---|
| Optimizer | AdamW | |
| Optimizer Momentum | $\beta_1, \beta_2 = 0.9, 0.999$ | |
| LR Schedule | Linear warm up cosine decay | |
| Warmup steps | $1e4$ | |
| Minimum LR Scale | 10 | |
| Gradient Clip Value | 1 | |
| Weight Decay | 0.01 | |
| Context Length | 16 | 256 |
| Encoder | SD-XL VAE Rombach et al. (2022); Stability | - |
| Image Dimension | 224x224 | - |
| Tokenizer | - | EleutherAI/gpt-neox-20b Black et al. (2022a) |
| Vocab Size | - | 50277 |

All models were trained using the large model size described in Table D.2, with a learning rate of $1e-4$ for $100,000$ steps. For the DiT baseline, we used the same hyperparameters from Peebles & Xie (2023), changing only the batch size to 128 from 256. We based our bidirectional EBT implementation on the code from this repository. We experimented with several different diffusion inference strategies to ensure fair comparison, including DDPM, DDIM, increasing the number of diffusion steps at inference, as well as recursing the diffusion model on its own denoised output. Ultimately, we found that the combination of DDIM recursed on its own output performed best, hence we used this as the baseline in all experiments. We used the default denoising schedule from the DiT codebase Peebles & Xie (2023). As the noise level $\beta$ was set to $0.1$ during training, and the default number of diffusion denoising steps was $1,000$, the number of denoising steps the diffusion model was trained on was $100$.

To make the denoising experiments compatible with diffusion models, we deviate from the original noising schemes performed in the denoising works mentioned in the main paper Du et al. (2022), and use a scheme based on the noising schedule from diffusion models. Specifically, we follow Peebles & Xie (2023), and use a linear variance schedule ranging from $1 \times 10^{-4}$ to $2 \times 10^{-2}$. To control the noise level, we use a hyperparameter denoted $\beta$ representing the percentage of the diffusion schedule to noise samples; $\beta$ was set to $0.1$ during training.

For both DiTs and EBTs, we found that models performed best on OOD noise levels when denoising their own outputs twice, that is applying the model to denoise the same image three times recursively. This is how we are able to get the results in Figure B.6 demonstrating the performance for 300 forward passes from DiTs and 3 forward passes from EBTs. We found that for image denoising, it was not necessary to train EBTs with the S2 hyperparameters for System 2 Capabilities to emerge, although its possible these would further improve performance. Additionally, for image classification, for both models, we take the average of all the final patch tokens, and for DiTs we feed in $T = 0$.

## D.4 COMPUTATIONAL RESOURCES

All experiments were conducted on either Nvidia A100s or GH200s, with the largest scale experiment requiring approximately $\approx 1300$ A100 GPU Hours. The runtime for each experiment was dependent on the model sizes used as well as the amount of data trained on.

## D.5 THEORETICAL FLOP AND MEMORY CALCULATIONS

We adopt the standard estimate of $6N$ FLOPs per token for the AR Transformer++ Casson (2023), where $N$ denotes the number of non-embedding parameters. For AR EBTs, however, the per-token cost varies with the number of training optimization steps and chosen hyperparameters.

To derive the FLOPs for EBT training, we follow Dagréou et al. (2024) for the Hessian-vector product (HVP), which EBTs require to backpropagate through a first-order derivative. Since an HVP has the same theoretical complexity as a gradient computation Pearlmutter (1994), we express the per-step

Table D.4: **Hyperparameters for EBT experiments.**

| Hyperparameter | CV | NLP |
|---|:---:|:---:|
| Optimizer | AdamW | |
| Optimizer Momentum | $\beta_1, \beta_2 = 0.9, 0.999$ | |
| LR Schedule | Linear warm up cosine decay | |
| Warmup Steps | $1e4$ | |
| Minimum LR Scale | 10 | |
| Gradient Clip Value | 1 | |
| Weight Decay | 0.01 | |
| Context Length | 16 | 256 |
| Encoder | SD-XL VAE Rombach et al. (2022); Stability | - |
| Image Dimension | 224x224 | |
| Tokenizer | - | EleutherAI/gpt-neox-20b Black et al. (2022a) |
| Vocab Size | - | 50277 |
| Optimization Steps | 2 | 2 |
| Optimization Step Size | $30,000$ | 500 |
| Optimization Step Size LR Multiplier | $90,000$ | $1,500$ |
| Learnable Optimization Step Size | ✓ | |

FLOPs as

$$\text{FLOPs} = F + B + B.$$

Based on Casson (2023), the forward and backward passes require approximately $2N$ and $4N$ FLOPs per token, respectively, where $N$ denotes the count of non-embedding parameters in the Transformer. In the autoregressive EBT implementation, the effective sequence length becomes twice that of the original Transformer (formally $2S - 2$ for an original sequence length $S$). Owing to the efficient scheme of Section C.6, this doubling of sequence length translates roughly into a two-fold increase in FLOPs, rather than a four-fold increase in FLOPs. Hence, each second-order optimization step demands roughly

$$(F + B + B) \times 2 = (2N + 4N + 4N) \times 2 = 10N \times 2,$$

making it $\approx 3.33\times$ more expensive than a standard feed-forward Transformer step.

The overall FLOP count also depends on one's choice of hyperparameters. For S2 models, where the loss is evaluated at every iteration without gradient truncation, the total FLOPs simply multiply by the number of steps. Therefore, for our pretraining experiments using two steps, we get that EBTs used $6.66\times$ the FLOPs of a comparable Transformer++ training. In contrast, for S1 models, a random number of optimization steps are used, the gradient is truncated, the loss is only calculated at the last step following Du et al. (2022), and a Replay Buffer is used. Therefore, the FLOP count varies and can both decrease (as truncating uses less FLOPs for earlier steps) as well as increase (as using more steps and a replay buffer both use more FLOPs). These numbers also vary during inference, where the full EBT implementation parallelizing all predictions at once is not necessary.

Given the scarcity of published methods for computing higher-order derivative FLOPs and our inability to leverage existing libraries for Hessian-vector products, these estimates remain approximate. We welcome corrections or additional insights from readers familiar with FLOP calculations for second-order methods.

The theoretical memory utilization of EBTs compared to normal feed-forward Transformers follows the same trend and calculations, where optimal HVP computation involves linear time and memory the same as standard backpropagation Pearlmutter (1994).

### D.6 EMPIRICAL HARDWARE BENCHMARKING

In addition to discussing theoretical FLOPs, to help understand performance more practically with modern hardware and software, we discuss empirical hardware performance and benchmarks for EBTs versus the more standard Transformer++. We conduct experiments measuring throughput, memory usage, and latency of EBTs and Transformer++ Autoregressive Language Models. We measure performance three times for each metric, and report an average. All experiments are with

Table D.5: **Training Efficiency of XXS Models.** Throughput (iterations/s after 5 minutes), memory usage (GB, batch size 32), and latency of the first training step (ms) over three runs and their average.

| Model | Trial 1 | Trial 2 | Trial 3 | Avg |
|---|---|---|---|---|
| **Throughput (it/s, higher is better)** | | | | |
| EBT 1 MCMC (xxs) | 4.35 | 4.28 | 4.54 | 4.39 |
| EBT 2 MCMC (xxs) | 2.18 | 2.16 | 2.30 | 2.21 |
| Base (xxs) | 16.74 | 16.75 | 18.16 | 17.2 |
| **Memory (GB with batch size 32, lower is better)** | | | | |
| EBT 1 MCMC (xxs) | 39.6 | 39.6 | 35.7 | 38.3 |
| EBT 2 MCMC (xxs) | 62.6 | 62.6 | 59.7 | 61.6 |
| Base (xxs) | 22.4 | 22.4 | 19.9 | 21.6 |
| **Latency for first training step (ms, lower is better)** | | | | |
| EBT 1 MCMC (xxs) | 1020 | 1020 | 1020 | 1020 |
| EBT 2 MCMC (xxs) | 1860 | 1840 | 1900 | 1870 |
| Base (xxs) | 587 | 592 | 552 | 577 |

XXS models using a batch size of 32, and memory/throughput being reported after 5 minutes of training. We measure EBTs with both 1 optimization step and 2 optimization steps to determine the impact of the number of steps on performance The results are shown in Table D.5, where the empirical performance of EBTs compared to Transformers very roughly matches theoretical FLOPs and memory. It's possible that a large part of the mismatch stems from PyTorch implementing HVPs suboptimally, with Jax implementing them much more efficiently Dagréou et al. (2024). Future algorithmic progress of EBTs could drive further PyTorch software development with better support for efficient HVPs.

## D.7 DATA-CONSTRAINED EXPERIMENTS

All models are trained for up to $100k$ training steps, where we often start to observe overfitting within the first $10k$ training steps. We use a learning rate of $1e-4$ for all models, a batch size of $64$, and a weight decay of $0.01$. For RNNs we experiment with the number of iterations being in the range of $5-20$, and for a fair comparison, we experiment with EBTs having the number of MCMC steps in the range of 2 to 7 (which is actually less compute per training step than the RNNs). Both RNNs and EBTs were only trained at the XXS scale. Because feed-forward Transformers have no notion of inference-time computation, we also ablated training larger Transformer models to favor them (small, instead of just XXS), where performance improved from $0.0\%$ to $0.03\%$ moving from XXS to Small. Small transformers use more FLOPs per forward pass than the XXS EBTs and RNNs that achieved much better test-set performance, revealing how even when FLOP-matched, Transformers perform worse than EBTs/RNNs due to overfitting data rather than learning generalizable reasoning.

## E RELATED WORKS

### E.1 TRADITIONAL TRANSFORMERS

The Transformer architecture Vaswani et al. (2017) has become ubiquitous across various domains Latif et al. (2023); Oquab et al. (2023); Radford et al. (2019); Touvron et al. (2023). The most commonly used transformer variant of today makes predictions directly in the output space with a single forward pass, demonstrated in Figure 1a. Because these models have a finite depth and width, and make predictions in a single forward pass, they are unable to dynamically allocate more computation to *each* prediction being made. Furthermore, they cannot model uncertainty in continuous state spaces in the same way they can in discrete state spaces because the normalization process for continuous state spaces is not as well-defined as it is for discrete spaces using softmax Dawid & LeCun (2024). Rather, training these models to express uncertainty relies on tricks such as Vector Quantization Van Den Oord et al. (2017) or pseudo losses/objectives (e.g., ELBO Kingma

et al. (2013)). Finally, because these models are not trained to explicitly verify samples, improving inference-time performance at a per-prediction level often requires external models Lightman et al. (2023).

## E.2 RNNs

Recently, several RNN variants (Figure 1b) have emerged to alleviate memory bottlenecks and achieve faster inference Gu & Dao (2023); Peng et al. (2023). These approaches have scaled similarly to Transformers in autoregressive sequence modeling and achieve better memory efficiency and reduced latency. However, traditional RNNs updating internal state based only on new information/data Gu & Dao (2023); Peng et al. (2023) are not capable of allocating additional computation during inference, and thus suffer from the same flaws as traditional transformers in achieving human-like System 2 Thinking.

To resolve these issues, people have equipped RNNs with the ability to allocate computation dynamically, with architectures such as the Universal Transformer Dehghani et al. (2018). Recently, this type of RNN has also been applied to LLMs Geiping et al. (2025); Saunshi et al. (2025), allowing LLMs to reason using additional computation in a continuous latent space through the depth of an unrolled RNN. Parallel work has developed RNNs for algorithmic reasoning Jolicoeur-Martineau (2025); Wang et al. (2025). However, like Diffusion models, these models learn to amortize gradient prediction of the energy function Geiping et al. (2025), rather than learning to explicitly verify predictions, meaning they cannot model uncertainty or explicitly verify predictions. Consequently, EBMs generalize these RNN-based architectures by offering explicit prediction verification capabilities Ma et al. (2025). Further discussion on this relationship is provided in Section E.6.

## E.3 DYNAMIC COMPUTATION (THINKING) WITH LLMS

The ability to leverage a dynamic amount of computation in LLMs has been emulated using chain-of-thought prompting Wei et al. (2022) and continuous latent space reasoning Hao et al. (2024). While these approaches can improve performance, they don't seamlessly transfer to continuous modalities, and LLM chain-of-thought has been shown to be unreliable for reasoning Agarwal et al. (2024); Lin et al. (2025); Turpin et al. (2023). More recently, models have been explicitly trained to perform reasoning using Reinforcement Learning Anthropic (2025); Guo et al. (2025); Jaech et al. (2024); xAI (2025). These approaches allow LLMs to simulate additional computational depth based on the number of tokens decoded before making a prediction, and as a result, significantly improve performance Guo et al. (2025); Jaech et al. (2024). The main limitations of these approaches are that they currently apply only to discrete domains (i.e., LLMs), are effective on a narrow set of problems that are easily verifiable (e.g., math and coding), and require additional supervision, typically in the form of reward signals, making them incompatible with purely unsupervised pretraining Guo et al. (2025).

## E.4 DYNAMIC COMPUTATION (THINKING) WITH DIFFUSION

The most common instance of a model architecture specifically created to leverage dynamic computation is diffusion models (Figure 1c), where using multiple forward passes to generate a prediction is a core aspect of both training and inference Höppe et al. (2022); Rombach et al. (2022). Although diffusion models implicitly define a likelihood through the reverse process Ho et al. (2020), which could theoretically be used to verify predictions, in practice an external verifier is necessary to improve performance at inference time beyond increasing denoising steps Liu et al. (2025); Ma et al. (2025); Singhal et al. (2025). This requirement limits the generalizability and scalability of diffusion models as an approach for System 2 Thinking, as they do not have two of the cognitive facets described: the ability to model uncertainty in continuous state spaces or explicitly verify predictions without additional models Liu et al. (2025); Ma et al. (2025); Singhal et al. (2025). Furthermore, diffusion models rely on a fixed denoising schedule, which restricts their ability to adaptively halt or extend computation—unlike EBMs. Additionally, diffusion models can be seen as predicting the gradient of the data density/energy function Du et al. (2023), and therefore that EBMs are a generalization of diffusion models that learn to explicitly verify predictions. More on this connection is in Section E.6, and a side-by-side comparison of diffusion models and EBMs is in Figure E.1.

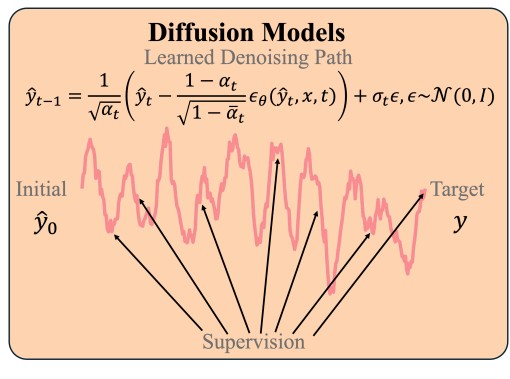 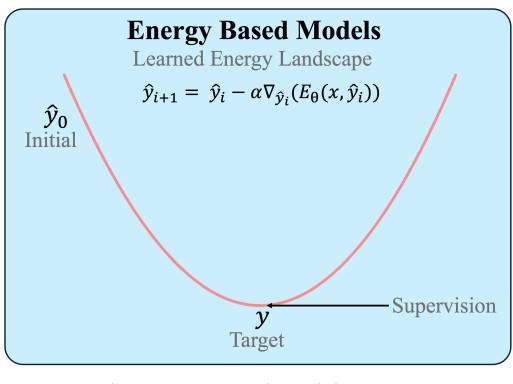

(a) Diffusion Model    (b) Energy-Based Model (EBM)

Figure E.1: **EBM and Diffusion Comparison.** Diffusion models receive supervision at each step of the denoising process (e.g., for one thousand steps), whereas EBMs only receive supervision at the end of the optimization process. This training procedure allows EBMs to learn an entire Energy Landscape over predictions, associating a scalar energy for every prediction according to its likelihood. Learning landscapes in this manner can reduce "error" accumulation throughout the denoising process Du et al. (2024) and makes EBMs more flexible by allowing unnormalized likelihood estimation at each step of the denoising process. Additionally, diffusion models update predictions by predicting the noise at each timestep, meaning they must follow a set denoising schedule. On the other hand, EBMs update predictions by performing gradient descent with respect to the energy scalar, allowing for flexible inference where this optimization process can be performed for any number of steps. $x$ here refers to some condition (e.g., a class or text) whereas $y$ is the generated prediction.

### E.5 ENERGY-BASED MODELS (EBMS)

The perspective of energy minimization as thinking/reasoning has been known for some time LeCun et al. (2006). Therefore, the most similar approaches to EBTs also train EBMs to do reasoning/thinking Du et al. (2022; 2024). While these works achieved impressive generalization results, they only focus on small-scale problems, and did not scale EBMs to high-dimensional real-world problems such as language or video. Additionally, these works did not perform an in-depth analysis on the types of System 2 Thinking that emerge with EBMs, more complex inference time procedures beyond just increasing the number of gradient descent steps, approaches towards improving EBM scalability, and required techniques for enhancing System 2 Thinking in EBMs.

### E.6 ENERGY-BASED MODELS (EBMS) AS A GENERALIZATION OF DIFFUSION MODELS AND RECURRENT DEPTH MODELS

Both diffusion models Du et al. (2023) and RNNs Saunshi et al. (2025) can be seen as predicting the score, or the gradient of the energy function/data density, $\nabla_x E_\theta(x)$, where diffusion models do this with an additional time condition Liu et al. (2022). Thus, the largest benefit of explicit EBMs over these approaches, which can be seen as implicit EBMs (due to only implicitly defining an energy function), is that using an explicit EBM allows for explicit verification/likelihood modeling. We show that this enables the use of self-verification to improve predictions, whereas with diffusion models and RNNs an additional verifier model is necessary to achieve this capability Ma et al. (2025).

It's also worth noting that RNN, diffusion models, and EBMs need not be incompatible with one another. For example, Du et al. (2024) combines EBMs and diffusion to reason over challenging problems. This can increase stability of the learned energy landscape by adding explicit score supervision.

### E.7 EBM AND DIFFUSION MODEL COMPARISON

Because of the similarity of diffusion models and EBMs, we present a side-by-side comparison of the training and inference approach for both in Figure E.1, where the primary difference lie in the supervision they receive during training and the update rule for predictions. Additionally, we provide more information to compare these approaches.

Under the assumption that the energy landscape is well formed and that optimization is well behaved, EBMs offer several distinct advantages over diffusion models. In EBMs, the energy function is trained to represent a meaningful landscape where the energy value of a sample directly corresponds to its relative unnormalized likelihood. Consequently, two samples can be directly compared to determine which is more likely, in a single forward pass. On the other hand, diffusion models require running samples through the entire reverse diffusion process to get likelihoods, which often requires hundreds to thousands of steps, and rely on likelihood approximations such as ELBOs or numerical solvers for SDEs/ODEs. In practice, these result in incomparable likelihoods, as ELBOs only give likelihood lower bounds and numerical solvers result in high approximation error Ma et al. (2025).

Furthermore, the learning of an energy landscape means that any approximation errors at each individual step of the Markov Chain (optimization process) do not result in cumulative error, as the minimum of the energy landscape can still be reached. This differs from diffusion models, where any approximation error at each step will result in increasing accumulated error across the entire Markov Chain Du et al. (2024) (demonstrated in Figure E.1).

Lastly, EBMs, giving an unnormalized likelihood estimate at each step, are in practice much more flexible for generation than diffusion models. While diffusion models require running the entire reverse diffusion process with a specific denoising schedule to generate a sample, EBMs can be trained to directly predict the next sample in a single step, and giving an unnormalized likelihood at each step can indicate how likely they think this sample is. This better approximates human-like System 2 thinking, where humans naturally evaluate the strength of current predictions, and on the basis of knowing how good their predictions are, decide to dynamically allocate more or less computational resources.

### E.8    ADDITIONAL ENERGY-BASED MODELS RELATED WORKS

One contribution of this work was the design of a custom architecture for EBM's called the Energy-Based Transformer (EBT). Roughly similar is the work of the Energy Transformer Hoover et al. (2024). Despite strong similarity in the names of these architectures, however, they are very different—with the primary similarity in architectures being the usage of attention mechanisms as well as a global energy function. The existing work integrated ideas from Modern Hopfield Networks, including RNNs, whereas in our work the architecture is non-recurrent and does not use associative memories. Additionally, EBTs differs with its focus on System 2 Thinking, which this previous work did not experiment with.

Other somewhat similar approaches to EBTs involve autoregressive Energy-Based Models, including E-ARM Wang et al. (2022), EBR Bhattacharyya et al. (2020), and Residual EBMs Bakhtin et al. (2021). E-ARM involves adding an objective to the learning process to turn a traditional autoregressive model into an EBM, and as such does not achieve two of the cognitive facets discussed. EBR and Residual EBMs involve the training of an EBM on top of an already pretrained autoregressive language model. Both works, however, leverage a contrastive objective, which suffers from the curse of dimensionality.

The optimization procedure used to train EBTs can be seen as a form of denoising score matching Vincent (2011); Wang et al. (2023). Particularly, predictions being initialized at some Gaussian, and then being optimized using the gradient of the energy function can be seen as training the EBM to denoise by learning the score of the data. However, we find the optimization perspective is more intuitive, and this denoising score matching perspective is more similar to the diffusion model training procedure than it is EBMs, involving multiple levels of noise rather than just one.

## F    ADDITIONAL COGNITIVE FACETS

**Facet 3: Modeling Uncertainty in Continuous State Spaces.** While thinking longer is important for improving performance, humans also weigh how uncertain they are before committing to a decision. In language, LLMs can simulate this through token-level probabilities Tomani et al. (2024). In the context of continuous state spaces, such as in vision, without the usage of discretization schemes such as Vector Quantization Van Den Oord et al. (2017) or pseudo losses/objectives (such as ELBO Kingma et al. (2013)), standard implementations of the most successfully used approaches with Transformers, RNNs, or Diffusion models generally do not provide strong or reliable uncertainty estimates Heng

et al. (2024); Nalisnick et al. (2018); Sankararaman et al. (2022); Serrà et al. (2019). [10] EBMs can naturally model uncertainty without having to model exact likelihoods Dawid & LeCun (2024) by modeling the relative unnormalized likelihoods of predictions, as demonstrated in Figure 3. As the real world often contains many inherently unpredictable elements, for instance, when a pedestrian might emerge from behind a parked vehicle, the ability to express uncertainty in predictions is essential to being cautious, and is a natural capability of humans Peters et al. (2017); Sarinopoulos et al. (2010); Vilares et al. (2012).

**Facet 4: Compositional Reasoning and Systematicity.** Humans routinely solve novel tasks by recombining familiar primitives (e.g., verbs with new arguments or visual parts into unseen objects), a hallmark of compositional generalization well-documented in neuroscience Friederici & Weissenborn (2007). In contrast, state-of-the-art Transformers and diffusion models often fall short when evaluated on compositional generalization Huang et al. (2023); Kobayashi et al. (2024). Energy-Based Models (EBMs) seamlessly address these limitations: energies for individual factors are composable in several different manners Du et al. (2023), enabling zero-shot generation of novel combinations without retraining, where gradient-based sampling provides an intrinsic mechanism to verify and iteratively correct compositions Du et al. (2023). Thus, EBMs offer a promising path toward human-like systematicity that remains elusive for existing approaches.

## G    COUNTERARGUMENTS

### G.1    SYSTEM 2 THINKING

In this paper, strong claims were made regarding the capabilities of current models and their ability to perform System 2 Thinking. However, there are common counterarguments to our claims, which we address here, in hopes of clarifying why we believe this is not currently possible.

#### G.1.1    SYSTEM 2 THINKING AND INFERENCE-TIME COMPUTE TERM USAGE

Whether the computational effort spent at inference time fully captures what psychologists term System 2 Thinking is still actively debated. In Section C.1 we outline three reasons for preferring the broader label System 2 Thinking: (i) it naturally extends to settings such as continual learning where terms such as "inference-time compute" becomes ambiguous, (ii) it connects our discussion to a substantial body of cognitive-science work, and (iii) it offers a conceptually straightforward entry point for readers beyond the machine-learning community.

It is widely acknowledged, and we agree with the idea, that human System 2 Thinking encompasses a far greater depth and complexity than the specific approaches explored in this paper, such as "Thinking Longer" and "Self-Verification." We wish to emphasize that our work does not claim current models replicate the full spectrum of human System 2 Thinking. Rather, we view the methods presented here as foundational steps toward that more ambitious long-term goal.

We propose that "System 2 Thinking" offers a useful umbrella term that can effectively encompass and generalize other existing terminologies, including "inference-time compute," "test-time compute," or "reasoning." A parallel can be drawn with the term "learning" in our field. "Learning" itself has evolved to describe a wide array of processes, some of which, such as k-Nearest Neighbors (KNNs) or the specific mechanisms of weight matrix updates in Artificial Neural Networks (ANNs), represent distinct facets rather than the entirety of human-like learning, meaning these approaches may not encompass the complexity of true human-like learning. However, despite this breadth and these simplicities, "learning" has become a cornerstone term, upon which our entire field of machine learning is named upon.

In a similar vein, while the "thinking" exhibited by the models discussed in this paper may not yet capture the full nuance and intricacy of human cognition, we believe the term "System 2 Thinking" can still serve a valuable role. It offers a generalizing framework for existing vocabulary and can contribute to making complex concepts within the field more accessible and understandable to

---

[10]We acknowledge that there are approaches to achieve uncertainty with the models discussed, such as Mixture Density Networks Bishop (1994) as well as score-based diffusion models Song et al. (2020). However, these approaches have seen less widespread success and scalability than the current dominant approaches.

newcomers or experts from other fields. Our intention is to contribute to a constructive and unifying dialogue of intelligent systems.

### G.1.2 IS CHAIN-OF-THOUGHT (COT) SUFFICIENT FOR SYSTEM 2 THINKING?

CoT is commonly thought to be sufficient for advanced reasoning to emerge in LLMs. However, in this paper we argue there are several flaws with CoT preventing advanced thinking capablities. First, Chain-of-Thought (CoT) involves reasoning over a discrete state space, which limits the granularity of "thoughts." Second, CoT is not an intrinsic architectural capability but an external procedure applied to token sequences. Ideally, such reasoning should be embedded within the model and learned during training. Third, each token is produced with a fixed computational budget, restricting the depth of reasoning per step. In contrast, humans allocate variable effort across steps when reasoning "step by step". Similarly, models should be able to spend a variable amount of computation per token, as enabled by EBTs. This aligns with the intuition behind the saying: "a chain is only as strong as its weakest link"—each step in the chain should receive sufficient computation to avoid failure points that result in bad reasoning chains.

## H    ENERGY-BASED MODELS (EBMS) INTRODUCTION

### H.1    SIMPLIFIED ENERGY-BASED MODEL (EBM) INTRODUCTION

Feed-forward neural networks generally take the form of: given an $x$ predict $y$ (Fig. H.1a). Energy-Based Models (EBMs) are a family of models that learn the compatibility (unnormalized probability) over all possible combinations of $x$ and $y$ (Fig. H.1b). Intuitively, this can be seen as learning to verify the strength of $y$ as a prediction with $x$ as the input. Training models in this manner allows for representing multiple plausible versions of $y$ compatible with a given $x$. The differences between these models is visualized in Fig. H.1.

Formulating models in this manner ultimately brings about two primary questions:

**First question:** Assuming we still care about ultimately predicting $\hat{y}$ *how do we use such an EBM to predict $\hat{y}$?* With feed forward models, generally we can just input $x$ and get the output of the model as $\hat{y}$, but we can't do this with EBMs?

With EBMs, what happens is conceptually similar to diffusion models Rombach et al. (2022), where we (commonly) initialize $\hat{y}$ as random noise. Then, we input $x$ and $\hat{y}$ into the model, and get a single scalar energy output (our initial energy output) from the model. Now, because our entire model is differentiable, we can get the *gradient from this energy scalar to $\hat{y}$* and perform gradient descent along the *energy landscape* (energy landscapes are surfaces resulting from mapping all possible predictions to scalar values, visualized in Figure 3) using this gradient (this is the key)! This process is visualized in Figures 3, 2. This gradient can be seen as the opposite of the noise (e.g., denoising) and therefore EBMs have strong relations with Diffusion models predicting the noise. EBMs can be seen as a generalization of diffusion models, where diffusion models are predicting the gradient of the energy function/scalar (more on this in Section E).

**Second question:** *How do we train an EBM?* Generally, models are trained over a dataset of $x$ and $y$ pairs, but now there are several different possible $y$ values that can be associated with any given $x$ value—so how does that work?

It turns out that all the training techniques for EBMs boil down to two main categories: contrastive and regularizing approaches Dawid & LeCun (2024).

Contrastive approaches are more common for EBMs and are easier to rationalize about due to their similarity to GAN discriminators. The idea behind contrastive approaches is to push down on the energy of positive samples (i.e., the true data), and to push up on the energy of negative samples. While these positive samples are easy to rationalize about, as they are just the true data, the difficulty of contrastive EBMs is finding negative samples. Several approach exist, such as GANs, which use a generator to amortize negative sample generation, or running MCMC (similar to optimization) for some time. However, as discussed in Section 3.1, such approaches don't scale well due to the curse of dimensionality.

Therefore, to achieve a scalable EBM approach, we train EBMs through an optimization procedure (which has strong resemblance to Langevin Dynamics). That is, EBMs are trained to, starting from an initial prediction, optimize predictions to the ground truth solution (shown in Figures 3, 2). This pushes the energy landscape to be smooth with a single local minimum at the ground truth solution, thereby regularizing the energy landscape to have low energy only on the true data. As mentioned in Wang et al. (2023), this can be seen as being similar to denoising score matching Vincent (2011).

Commonly, when people learn about EBMs and the iterative denoising/optimization procedure performed during training, they think of diffusion models, so we include a more in depth comparison between the two in Section E.7.

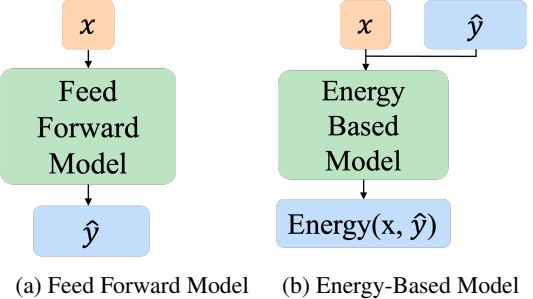

(a) Feed Forward Model    (b) Energy-Based Model

Figure H.1: **Feed-Forward and Energy-Based Model Comparison.** Feed-forward models (a), given an input $x$, directly try to predict $\hat{y}$. Instead of just getting $x$ as an input, EBMs receive both $x$ and $\hat{y}$ as an input and learn the **compatibility** of all possible values of $\hat{y}$ with $x$ by outputting a scalar energy value for each combination. Low energy corresponds to high probability, and high energy to low probability. In practice, $\hat{y}$ is often initialized as random.

## H.2    ENERGY-BASED MODEL TYPES

Because EBMs are a broad modeling framework, and can generalize many existing approaches, we aim to provide precise language to distinguish EBM types. We broadly classify the EBMs described throughout this paper as **explicit EBMs**, meaning they explicitly define an energy function over inputs as the entire function being learned. In other words, explicit EBMs directly map all variables (inputs) to a single scalar energy as the output of the neural network. We define these in contrast to **implicit EBMs**, or EBMs where the energy function is not the learned model but rather some implicit definition of the learned model. For example, with diffusion models, the energy function is implicitly defined by the learned score network ($s_\theta(x, t)$) as the following:

$$\nabla_x E(x, t) = -s_\theta(x, t)$$
$$E(x, t) = -\int^x s_\theta(u, t)\, \mathrm{d}u \; + \; C(t)\,.$$

Other notable examples of implicit EBMs include Hopfield Networks Hopfield (1982), RNNs Geiping et al. (2025), and Boltzmann machines.

## H.3    ENERGY-BASED MODEL FREQUENTLY ASKED QUESTIONS (FAQ)

- **What is energy/compatibility, what does it represent, and how is it learned? What energy corresponds to what probability?**
  Energy is just a learned compatibility score between $x$ and $y$ (lower means more likely). The EBMs described in this paper learn it implicitly as described in Section 3.1 such that the true data (good pairs) have low energy and bad pairs (non-data) have higher energy. Probabilities follow:

$$p_\theta(x) = \frac{e^{-E_\theta(x,y)}}{Z(\theta)}$$
$$p_\theta(x, y) \propto e^{-E_\theta(x,y)}$$

so the energy $E$ is essentially the (unnormalized) negative log-likelihood up to an additive constant. The term compatibility is just a term used for intuition.

- **Does training EBMs require a full Hessian calculation?**
  No—the approach described in the paper only requires Hessian-vector products. That makes training only about a constant $1.66\times$ as expensive as a vanilla feed-forward model given everything else remains constant and you use a single step.

- **Why is low energy good (and high energy bad)? Why not just use probability?**
  Low energy is good because of the negative exponential. The reason we don't use probabilities is avoiding normalized probabilities makes the problem much more tractable in real-world high-dimensional continuous state spaces by removing the focus on explicit normalization via regularizing the partition function. EBMs come from a long line of work in statistical physics.

- **Is it okay that energies are unnormalized probabilities?**
  Yes, for most real-world applications, you only ever need sample **relative likelihood** comparison; it's significantly less common to need the exact likelihood of samples. An example of this is reward models, which can be seen as EBMs (just multiplying the reward by $-1$), where all that really matters is the relative reward for choosing which sample to use or which behavior to perform.

- **Is it fine to not do Maximum Likelihood Training?**
  Contrary to what your intuition may say, the answer is **yes**! For most real-world distributions, data lies completely concentrated on a very thin manifold with no defined distribution outside of this manifold. Thus, directly doing Maximum Likelihood Estimation (MLE) training would push EBMs to have low energy on the true data manifold and then infinite energy off that manifold (as the probability of such samples is 0). We don't want this as it would make the score (gradient of the energy function) undefined and the energy landscape untraversable—so not doing MLE makes the problem tractable.

# I  ENERGY-BASED TRANSFORMERS (EBTs) TUTORIAL

## I.1  IMPROVING STABILITY AND SCALABILITY

Energy-Based Models are notorious for instability during training Arbel et al. (2020); Du & Mordatch (2019); Du et al. (2020); Li et al. (2023). Therefore, we experiment with several different hyperparameters to increase the stability and scalability of EBTs and EBMs in general.

### I.1.1  OPTIMIZATION STEP SIZE AND STABILITY

We found that the step size for gradient descent updates of predictions ($\alpha$) was one of the primary factors affecting the stability of EBTs. Thus, for S1 models, we make the step size a learnable parameter (this is not the case for S2 models). We calculate its learning rate by multiplying the model's learning rate by the step size learning rate multiplier. We found that the values for the step size have a large effect on the magnitude of gradients generated for the optimization of predictions. This is because the step size is directly multiplied by the prediction gradients. Particularly, a smaller step size results in larger generated gradients, whereas a larger step size results in smaller gradients. Therefore, the step size needs to be tuned per modality, as the update required for predictions depends on data. It's also worth noting that we do not weight decay the step size in any of the models.

We also found that a relatively high optimization step size was necessary ($30,000$ for video and between $5$ and $500$ for text). Without a high optimization step size, gradient magnitudes continued to increase throughout training, resulting in unstable training dynamics.

### I.1.2  ARCHITECTURE STABILITY

For autoregressive models, we found that simply prepending a learnable "step" embedding to sequences significantly improved scalability and stability, especially for S1 models. This step embedding mapped a discrete step index (i.e., step 0, 1, etc.) of the current optimization step to an embedding the same dimension as the model's embedding. We believe this helped improve stability

by enabling the accumulation of attention mass, as well as enabling less steep energy landscapes conditioned on the optimization step.

Additionally, we experimented with adaptive layer normalization from the DiT architecture, but we found that the timestep embedding worked better. We also experimented with several different normalization and initialization approaches, where we found that the standard Llama2 Touvron et al. (2023) architecture and initialization worked best. This involves using RMSNorm, Xavier init Glorot & Bengio (2010), SwiGLU MLP Shazeer (2020), and RoPE Su et al. (2024).

### I.1.3 SYSTEM 2 THINKING HYPERPARAMETER STABILITY

For S2 models, we found certain hyperparameters to be essential for the stability and scalability of models. First, we found that using a lower number of optimization steps resulted in more stability, as using more optimization steps necessitates longer gradient chains. Additionally, we found that the strategy used for the randomization of $\alpha$ to be very important for stability. Particularly, we found that randomizing $\alpha$ with a single value for an entire batch resulted in issues with training convergence. We believe this is because using the same value for every element in the batch resulted in high-variance gradients. Thus, when randomizing $\alpha$ differently for every batch and sequence element, we found training convergence to be more stable. It's possible that randomizing the number of optimization steps would yield similar results in reducing gradient variance, however, we did not experiment with such a configuration.

### I.1.4 CLAMPING OPTIMIZATION GRADIENTS FOR STABILITY

We found that clamping gradients of the energy function with respect to predictions (or prediction update gradients) could help improve training stability at the cost of some slight reductions in convergence speed. We do not conduct any experiments in the paper with clamped prediction gradients as we found that the other hyperparameters were sufficient for stable training, however, it's a potentially useful trick worth discussing.

### I.1.5 NORMALIZING DATA FOR STABILITY

We found that normalizing/standardizing input data was crucial for the stability of EBTs. An example of this was in our NLP experiments, where we ran experiments with and without normalizing probability distributions. The experiments with unnormalized distributions often had extreme activations as well as large loss spikes, whereas the experiments with normalized distributions (by applying softmax) were stable.

### I.2 HOW TO TRAIN YOUR EBT

The hyperparameters used for training EBTs are *extremely* important and can often be highly sensitive towards performance, so here we offer a guide toward hyperparameter tuning. First, we recommend starting off with training S1-EBTs, which are the easiest and most stable variant of EBTs not designed for System 2 Thinking. Then, once S1-EBTs can be trained successfully and scaled, we recommend changing the hyperparameters gradually towards the hyperparameters used for S2 models (we say gradually here as occasionally these parameters can cause instability and require additional tuning).

When training S1 EBTs, we recommend tuning hyperparameters in the following order:

- First, tune standard hyperparameters such as Learning Rate (LR), batch size, etc. Having a high batch size helps with stability by making gradients less noisy (because you initialize predictions from random noise this makes gradients noisier).

- Second, start tuning S1-specific hyperparameters—primarily alpha and its LR multiplier (we recommend keeping its LR multiplier around $3x$ the value of alpha) and then tuning the number of optimization steps.

- Third, potentially tune whether the step size is learnable and try other EBT architectures (inspired by DiT Peebles & Xie (2023) we tried a time embedding as well as adaptive layer normalization).

Once you have tuned these, the model should be stable and fine for most use cases. At which point, if you are desiring System 2 capabilities, you can proceed to the S2 models I.3. Some potential metrics to monitor and look out for include the gradient magnitudes (if these increase too much or spike a lot that's a bad sign) and the gap between the initial and final energy after optimization (if this is too high or low it could be a sign your model's alpha value needs to be adjusted).

Following Wang et al. (2023), we give pseudocode for training EBTs in natural language for language modeling (Listing 1) as well as in computer vision for autoregressive video modeling (Listing 2). The pseudocode is primarily for S2 models without any energy landscape regularization techniques. The first primary design decision in the presented pseudocode is whether or not to detach predictions in between steps. Not detaching predictions in between steps allows for more "Thinking Time" before making predictions, but makes the gradient computation graph longer and therefore increases the likelihood of stability issues with gradients. Similarly, calculating the loss at every step versus solely the last step enables model to "think for longer" before needing to make accurate predictions, and therefore affects the "funnel-like structure[11]" and smoothness of the energy landscape. For S2 models, we found that **not** detaching between steps was best, and similarly that calculating the loss only at the last step was best. For S1 models, we found the opposite to be most stable. Generally, if one is calculating the loss only at the last step, then one should not detach between steps as it's best if the gradient propagates to previous steps in this case. For more details on these techniques, we refer the reader to the source code as well as Section I.3.

```python
# make sure to enable gradient tracking
with torch.set_grad_enabled(True):
    loss_fn = nn.CrossEntropyLoss(weight=None, ignore_index=
    tokenizer_pad_token_id)

    context_embeddings = self.embeddings(input_ids[:, :-1]) # B, S, D
    next_tokens = input_ids[:, 1:]
    next_embeddings = self.embeddings(next_tokens) # B, S, V; are just
    used for shaping next tensor
    predicted_distributions = torch.randn_like(next_embeddings) # B, S, V
    ; initialize predictions as random

    for _ in range(num_steps):
        # Can optionally detach predicted distributions so that no
    gradient flows through past steps
        # predicted_distributions = predicted_distributions.detach()

        predicted_embeddings = self.vocab_to_embed(softmax(
    predicted_distributions)) # B, S, D; need to proj. to embed space for
     transformer to work in, use linear layer, weighted sum, etc

        all_embeddings = torch.cat((context_embeddings,
    predicted_embeddings), dim=1) # B, 2S, D
        predicted_energies = self.transformer(all_embeddings) # B, S, 1;
    this returns only energies for the predicted_embeddings

        # Compute the gradient of predicted energies w.r.t. predicted
    distributions
        predicted_distributions_grad = torch.autograd.grad(
            predicted_energies.sum(),
            predicted_distributions,
            create_graph=True
        )[0] # B, S, V

        # Perform gradient descent w.r.t. the energy function where self.
    alpha is the optimization step size
        predicted_distributions = predicted_distributions - self.alpha *
    predicted_distributions_grad
```

---

[11]While one may be eager to classify these landscapes as convex, such properties are unlikely in high-dimensional spaces spanned by highly nonlinear neural networks.

```
    # Calculate cce loss based on predicted and ground truth
    distributions, optionally at each optimization step or only at the
    end
    cce_loss = loss_fn(predicted_distributions, next_tokens)
```

Listing 1: **Autoregressive Language Model Training Pseudocode in PyTorch**

```
# make sure to enable gradient tracking
with torch.set_grad_enabled(True):
    loss_fn = torch.nn.SmoothL1Loss(beta=1.0) # use whichever loss
    function desired

    context_embeddings = embeddings[:, :-1] # B, S, D
    next_embeddings = embeddings[:, 1:] # B, S, D
    predicted_embeddings = torch.randn_like(next_embeddings) # B, S, D;
    initialize predictions as random

    for _ in range(num_steps):
        # Can optionally detach embeddings so that no gradient flows
    through past steps
        # predicted_embeddings = predicted_embeddings.detach()

        all_embeddings = torch.cat((context_embeddings,
    predicted_embeddings), dim=1) # B, 2S, D
        predicted_energies = self.transformer(all_embeddings) # this
    returns only energies for the predicted_embeddings # B, S, 1

        # Compute the gradient of predicted energies w.r.t. predicted
    embeddings
        predicted_embeddings_grad = torch.autograd.grad(
            predicted_energies.sum(),
            predicted_embeddings,
            create_graph=True
        )[0] # B, S, D

        # Perform gradient descent w.r.t. the energy function where self.
    alpha is the optimization step size
        predicted_embeddings = predicted_embeddings - self.alpha *
    predicted_embeddings_grad

    # Calculate reconstruction loss based on predicted and ground truth
    embeddings, optionally at each optimization step or only at the end
    reconstruction_loss = loss_fn(predicted_embeddings, next_embeddings)
```

Listing 2: **Autoregressive Video Model Training Pseudocode in PyTorch**

## I.3   HOW TO THINK USING YOUR EBT

Once an S1 EBT has been trained, we recommend tuning the System 2 hyperparameters in the following manner:

- Remove detaching tensors between optimization steps and add loss truncation so the loss is only calculated at the final step.
- Then, tune alpha, as it's by far the most important EBT-specific hyperparameter. But, do not make it learnable.
- Next, tune the number of optimization steps, including potentially a minimum and maximum number when randomizing the number of steps.
- Then add a replay buffer, Langevin Dynamics, and eventually a randomized alpha (step size). Tune all of these in tandem while tweaking the earlier parameters (particularly alpha).

It's possible that randomizing the number of steps for each sample within a batch would work better (similar to randomizing the alpha value within a batch). Additionally, it's worth mentioning that all

optimization steps are performed along the same energy landscape (same time embedding condition), unlike with S1-EBTs using multiple time steps.

