# OpenReview forum: "Energy-Based Transformers are Scalable Learners and Thinkers"
_ICLR.cc/2026/Conference — ICLR 2026 Oral_

### Official Review · Reviewer_KJ9J · 2025-10-17

**Soundness:** 2
**Presentation:** 2
**Contribution:** 2
**Rating:** 2
**Confidence:** 3

**Summary:**

This paper presents Energy-Based Transformers (EBTs), a framework that reconceptualizes prediction as an optimization process rather than direct generation. Instead of emitting outputs (tokens, pixels, etc.) in a single forward pass, EBTs learn an energy function that evaluates the compatibility between inputs and candidate predictions. Inference then proceeds through gradient-based refinement toward lower-energy (more consistent) states.

**Strengths:**

The approach enables two appealing properties often associated with "System 2" reasoning: (i) adaptive computation, where more challenging inputs naturally require more optimization steps, and (ii) explicit self-evaluation, since energy levels can serve as confidence or verification signals.
Empirically, the paper demonstrates competitive performance across discrete and continuous domains and reports improved robustness under distribution shifts. The idea of learning to verify rather than learning to generate feels conceptually elegant, and the result ,especially the scaling behavior and cross-domain generality, suggest that EBTs may offer a promising direction for inference-time reasoning emerging directly from unsupervised pretraining.

**Weaknesses:**

Weaknesses

- Due to computational limitations, experiments are limited to models of up to 800M parameters, making claims about scaling advantages at the billion-parameter or foundation scale speculative. We can discount the requirement for experiments in billion/trillion parameter range, due to constraint of academic resources. However,  the projected breakeven point at which EBTs would become FLOP-efficient relative to Transformer++ is estimated at roughly 10⁵² total FLOPs, a scale that is effectively beyond practical reach. For context, even a state-of-the-art exascale system (≈10¹⁸ FLOPS) would require approximately 10³⁴ seconds of continuous computation to reach this point. Hypothetically distributing the workload across all 8 billion people on Earth, each operating a dedicated exascale supercomputer 24/7 (≈8×10²⁷ FLOPS combined), the breakeven computation would still demand about 1.25×10²⁴ seconds, equivalent to 3.96×10¹⁶ years, or nearly 400 trillion human lifetimes (assuming 100 years per lifetime) .Such an extreme requirement implies that the nominal scaling crossover occurs far beyond any conceivable computational regime. Consequently, the reported scaling advantage, while theoretically positive, lacks practical significance: in realistic operating regimes, EBTs remain orders of magnitude less compute-efficient than Transformers, making it difficult to justify claims of superior scalability.


- EBTs require roughly 10× more compute to achieve the same validation perplexity, calling into question their current efficiency. The suggestion that EBTs could surpass advanced Transformer variants remains speculative without stronger empirical backing. The argument that the slope "suggests" that EBTs would surpass transformer++ at high levels of compute, while correct, would benefit from more backing.
- The scaling law analysis lacks transparency: the procedure for estimating exponents and the dataset used for fitting are not described, making reproducibility difficult. Including a detailed table of fitted data points would improve credibility.
- The iterative inference procedure introduces multiple hyperparameters (e.g., step count, learning rate, noise levels) that require tuning. This makes deployment and fair comparison more involved than with standard feedforward Transformers.


- The method incurs substantially higher computational cost per token, yet the comparisons are not normalized for equal FLOPs. Without such baselines, it is unclear whether the observed gains stem from architectural benefits or simply increased compute. For text generation in particular, where predictions are locally conditioned, the incremental benefit appears modest relative to diffusion-style alternatives.


- The reported “2.91% faster scaling” claim (highlighted in Figure 5) lacks methodological clarity, no explanation is given for how this value was computed or whether variance in scaling fits was accounted for.


- It is not specified whether models in Table 3 were compared under matched computational budgets. Clarifying this would improve interpretability of the downstream comparisons.

**Questions:**

------
Some additional questions:
- Thinking longer presents an interesting argument. I know that some prior ideas like "HRM/TRM", and "Scaling up Test-Time Compute with Latent Reasoning: A Recurrent Depth Approach", have explored similar ideas (thinking longer, although they don't have explicit verification mechanism).-
- It would be great to see where EBTs perform well in CONSTRAINED data settings, for eg, solving extreme-sized sudoku puzzles, mazes, and unsupervised object discovery.

---

> ### Author Response · Authors · 2025-11-21
> **Response to Reviewer KJ9J (Part 1/3)**
>
> We thank the reviewer for great experimental suggestions and kind comments. We are happy that the reviewer believes that EBTs demonstrate competitive performance across modalities, finds that EBTs demonstrate improved robustness, finds the core idea conceptually elegant, and believes that the scaling behavior suggests EBTs are promising for inference-time reasoning.
>
> > Due to computational limitations, experiments are limited to models of up to 800M parameters, making claims about scaling advantages at the billion-parameter or foundation scale speculative. We can discount the requirement for experiments in billion/trillion parameter range, due to constraint of academic resources. However, the projected breakeven point at which EBTs would become FLOP-efficient relative to Transformer++ is estimated at roughly 10⁵² total FLOPs, a scale that is effectively beyond practical reach. For context, even a state-of-the-art exascale system (≈10¹⁸ FLOPS) would require approximately 10³⁴ seconds of continuous computation to reach this point. Hypothetically distributing the workload across all 8 billion people on Earth, each operating a dedicated exascale supercomputer 24/7 (≈8×10²⁷ FLOPS combined), the breakeven computation would still demand about 1.25×10²⁴ seconds, equivalent to 3.96×10¹⁶ years, or nearly 400 trillion human lifetimes (assuming 100 years per lifetime) .Such an extreme requirement implies that the nominal scaling crossover occurs far beyond any conceivable computational regime. Consequently, the reported scaling advantage, while theoretically positive, lacks practical significance: in realistic operating regimes, EBTs remain orders of magnitude less compute-efficient than Transformers, making it difficult to justify claims of superior scalability.
>
> > EBTs require roughly 10× more compute to achieve the same validation perplexity, calling into question their current efficiency. The suggestion that EBTs could surpass advanced Transformer variants remains speculative without stronger empirical backing. The argument that the slope "suggests" that EBTs would surpass transformer++ at high levels of compute, while correct, would benefit from more backing.
>
> **Response:** Thanks for the great feedback, we sincerely apologize for the lack of details regarding these scaling laws. We took these to heart and revised our scaling experiments with more seeds and better parameters to make our experiments more comprehensive and better aligned with the literature.
>
> We kindly refer the reviewer to the [main rebuttal comment](https://openreview.net/forum?id=ZBj3Qp1bYg&noteId=VbcSJxquF6), which we believe explains our changes, with a key portion quoted below:
>
> “we conducted a new set of primary scaling law experiments (Figures 5a and 5b) with revised scaling parameters to improve the reliability and strength of these results. Although the initial manuscript contained details on these experiments in Section D.1.1, we have updated the manuscript to include more comprehensive details, and have updated Figures 5a and 5b with the newest results. For these experiments, we have decided to use scaling parameters more aligned with the literature, where we follow Chinchilla [1] scaling to scale model sizes and data proportionally (earlier experiments did not scale model sizes and data proportionally, and instead scaled data sublinearly). We have rerun these experiments with multiple random seeds, computed an $R^2$ value to determine significance, and have reported tables with our results, where we find that EBTs have a scaling rate 8.97% higher than the Transformer++ for language modeling (which is higher than the 2.91% before). While we are limited by computation from conducting much larger scale experiments (e.g. billions of parameters), we believe that results being more significant, holding across seeds, and using more common scaling parameters substantiates them. We hope that these experiments have addressed most comments regarding scalability and thoroughness of scaling law experiments, and we have updated claims to better reflect that we are unsure what large-scale performance will be with billions of parameters.”
>
> With these new results, the crossing point for EBTs and Transformers is computed to be around $1.7 \times 10^{30}$ FLOPs instead of the $10^{52}$ from before. As modern foundation models hover in the range of $10^{25}$ to $10^{27}$ FLOPs, this result is a lot more practically achievable in the coming years as computation continues to grow.
>
> We hope that these scaling experiments, with several random seeds as well as being more aligned with the scaling laws literature, help to reduce uncertainty regarding the performance of EBTs. We include more details in Section D.1.1 on the values used for the scaling laws, the coefficient computation, the $R^2$ values, as well as the procedure for estimating coefficients.

---

> ### Author Response · Authors · 2025-11-21
> **Response to Reviewer KJ9J (Part 2/3)**
>
> We sincerely thank the reviewer for inspiring us to improve these experiments. We strongly believe that future works can make EBTs scale even better or be more computationally efficient, just as several works have gone into improving the standard Transformer++ and DiT recipes.
>
> > The scaling law analysis lacks transparency: the procedure for estimating exponents and the dataset used for fitting are not described, making reproducibility difficult. Including a detailed table of fitted data points would improve credibility.
>
> > The reported “2.91% faster scaling” claim (highlighted in Figure 5) lacks methodological clarity, no explanation is given for how this value was computed or whether variance in scaling fits was accounted for.
>
> **Response:** We have added a detailed table including the fitted data points as well as the dataset and procedure for fitting the scaling law in Section D.1.1, we include an excerpt of the text below:
>
> “Following Chinchilla [1], we model the scaling behavior as a power law $L(C) = \beta C^{-\alpha} + E$, where $L$ is the loss, $C$ is the compute budget (FLOPs), and $E$ is the irreducible entropy of natural text. Assuming $E$ is constant across models, we approximate the scaling rate by fitting a linear regression in the log-log regime:
> $$ \log(L - E) \approx -\alpha \log(C) + \log(\beta) $$
> Here, $\alpha$ represents the scaling exponent. By comparing the exponents for EBTs and Transformers, we can determine which architecture improves loss more efficiently per unit of compute.
>
> Based on the fit linear regression, we also compute an $R^2$ coefficient to determine the reliability of these scaling laws for FLOPs, where we get a value of $99.69\%$ for the Transformer++ and a value of $98.40\%$ for EBTs for FLOPs, both of which are within well-accepted ranges for Chinchilla scaling laws [2].”
>
> We believe that the added seeds and the computed $R^2$ computation accounts for most of the variance in scaling laws, and are happy to include any additional details or experiments the reviewer suggests.
>
> [1] https://arxiv.org/pdf/2203.15556
> [2] https://arxiv.org/pdf/2406.18922
>
> > The iterative inference procedure introduces multiple hyperparameters (e.g., step count, learning rate, noise levels) that require tuning. This makes deployment and fair comparison more involved than with standard feedforward Transformers.
>
> **Response:** We agree that introducing multiple hyperparameters makes deployment and comparisons more involved than standard feedforward Transformers. However, EBTs naturally expose more knobs that make them more flexible during training and inference, enabling better adaptation to different deployment constraints and objectives. As industry has headed towards large labs investing substantial compute and effort into hyperparameter tuning for foundation models, we believe this added flexibility is a feature rather than a bug, as it expands the design space teams can exploit and enables higher flexibility during training and inference. In section I.2 we include guidance on how to train EBTs, including how to tune hyperparameters, and provide strong default hyperparameter values in the code.
>
> > For text generation in particular, where predictions are locally conditioned, the incremental benefit appears modest relative to diffusion-style alternatives.
>
> **Response:** We generally agree that local conditioning (i.e., energy landscapes for each token) is worse than having an energy landscape for all tokens. We would like to clarify that we do conduct experiments with global energy landscapes (for the denoising experiments), and obtained strong results (better OOD generalization than diffusion models, 99% faster denoising). However, while experimenting with EBTs operating over entire sequences on data with many modes, i.e. the mentioned failed text-to-image experiments in Section B.3, we experienced poor generation quality/blurry images/mode collapse. We believe this is an issue with the current optimization formulation, which is unable to deal with a lot of uncertainty and plan to explore this issue as a promising future work.
>
> We briefly mentioned this challenge with capturing many modes in Future works under Section A.9, and have updated text to highlight how important a direction this is. We have also added this as a limitation to be more explicit and upfront about it.
>
> We still believe that verification happening at the per-token level, such as with the AR EBTs for language generation, are valuable as they enable making sure future states are compatible with the given sequence. However, in the long-term, we agree with the reviewer’s sentiment regarding EBTs operating over entire sequences as in Diffusion, and will work hard to realize this potential in future work.

---

> ### Author Response · Authors · 2025-11-21
> **Response to Reviewer KJ9J (Part 3/3)**
>
> > It is not specified whether models in Table 3 were compared under matched computational budgets. Clarifying this would improve interpretability of the downstream comparisons.
>
> **Response:** We have modified the caption in Table 3 regarding the computational budgets, with the goal of clarifying the setup as well as the purpose of those experiments, as follows:
>
> “We conduct experiments aimed at demonstrating the generalization of EBTs. Despite having slightly higher pretraining perplexity, EBTs often achieve lower perplexity on downstream tasks than the Transformer++, indicating better generalization. All models are trained with the same amount of data and parameters, but because EBTs at the current scale are less FLOP efficient (see Figure 5b), they used more FLOPs for this experiment. BB stands for BigBench.”
>
> Please let us know if anything is unclear or if any other modifications are requested.
>
> > It would be great to see where EBTs perform well in CONSTRAINED data settings, for eg, solving extreme-sized sudoku puzzles, mazes, and unsupervised object discovery.
>
> > The method incurs substantially higher computational cost per token, yet the comparisons are not normalized for equal FLOPs. Without such baselines, it is unclear whether the observed gains stem from architectural benefits or simply increased compute.
>
> **Response:** We agree that data-constrained settings are particularly interesting–we have conducted a set of experiments on sudoku puzzles. We compared a standard feed-forward Transformer, an RNN Transformer (similar to TRMs [1]]), and EBTs. Our results (described in the Experimentation Section) are strong with EBTs scoring 29.7% on the out-of-distribution test set, RNNs scoring 17.7% on the test set, and standard feed-forward Transformers scoring 0.03%--suggesting EBTs are able to generalize better due to their reasoning capabilities. We believe EBTs outperform other approaches because the verification capabilities are well suited for reasoning tasks, such as Sudoku.
>
> | Model         | OOD Test-Set Performance (%) |
> | ------------- | ---------------------------- |
> | Transformer++ | 0.03                          |
> | RNN           | 17.7                         |
> | EBT           | **29.7**                         |
>
> These results make it clearer that the observed gains are indeed due to EBTs generalizing better, and not due to leveraging more computation.
>
> To confirm further that the OOD test-set performance is due to improved generalization and not just more computation, we conducted an ablation where we tested Small Transformer++ models to XXS RNN/EBT models. The results, described in Section D.7, show that even using a Small Transformer++ model with more FLOPs/parameter per forward pass than RNNs/EBTs does not yield much of an improved performance (0.0% vs 0.03%), and it outperformed significantly by EBTs and RNNs.
>
> > Thinking longer presents an interesting argument. I know that some prior ideas like "HRM/TRM", and "Scaling up Test-Time Compute with Latent Reasoning: A Recurrent Depth Approach", have explored similar ideas (thinking longer, although they don't have explicit verification mechanism).-
>
> **Response:** Thanks for mentioning these works. These approaches are promising, as similar to EBTs, they enable more dynamic test-time compute than standard feed-forward models. As you have mentioned, we agree that the verification mechanism EBTs offer is one primary advantage, which could principally be used to determine when to stop/continue reasoning. In the data-constrained experiments we added, we compared EBTs to RNNs that operate very similarly to TRM, and we observed that EBTs do indeed generalize better, as shown in Table 5. An interesting future direction would be to compare EBTs on tasks such as ARC-AGI more thoroughly that these existing approaches (i.e. HRM, TRM) were more specialized for. We have added these works in our related works section.

---

> > ### Comment · Reviewer_KJ9J · 2025-11-23
> > **Thanks for the rebuttal**
> >
> > I thank the authors for their detailed rebuttal, my concerns are addressed, and seeing the strong ratings from other reviewers, i will increase my score to 6. I look forward to learning more about this paper at ICLR.

---

> > > ### Author Response · Authors · 2025-11-24
> > >
> > > Thank you for your thoughtful review and kind response. We appreciate your reconsideration of the score and are glad our responses addressed your concerns.

---

> ### Comment · Reviewer_KJ9J · 2025-11-25
>
> You are most welcome :-)

---

### Official Review · Reviewer_8YzV · 2025-10-23

**Soundness:** 4
**Presentation:** 3
**Contribution:** 4
**Rating:** 8
**Confidence:** 3

**Summary:**

This paper proposes EBTs that learn to verify the compatibility between inputs and candidate predictions through energy minimization. EBTs address limitations of existing System 2 Thinking approaches by enabling dynamic computation allocation and prediction verification across modalities without requiring additional supervision. The work introduces techniques for stable and parallelizable training of EBMs, demonstrating improved performance over Transformer++ and DiTs on language modeling and image denoising tasks while using fewer forward passes.

**Strengths:**

- This paper is well written with clear motivation analysis, comprehensive experimental design, and informative figures that effectively support the main arguments.
- The core insight of building generation models based on the principle that verification is easier than generation is novel and insightful. This approach provides a fresh perspective on how to design generative models by leveraging verification capabilities.
- The dynamic computation allocation based on problem difficulty is an interesting contribution that offers an alternative technical pathway for adaptive computation. This differs from common approaches that rely on layer skipping and provides a new way to handle varying computational requirements.
- The experiments are comprehensive, particularly in the language modeling domain where multiple scaling dimensions are explored to validate the proposed method. The extensive evaluation across different model sizes, data scales, and computational budgets demonstrates the robustness and effectiveness of the approach.

**Weaknesses:**

- The verification approach in this work does not align with intuitive expectations. For language tasks, meaningful verification should operate on complete sentences or full responses to questions. However, EBT uses autoregressive generation where each token is verified independently during generation. This raises questions about the validity of verification at the token level rather than at the semantic level of complete thoughts or responses.

- This work lacks sufficient analysis of computational efficiency. The paper primarily uses forward pass counts as an indirect measure, but absolute metrics such as latency or other time-based measurements would provide a more objective assessment of EBT's computational performance compared to Transformer++. Such analysis is crucial for understanding the practical viability of EBT.

- The implementation details of EBT are not clearly explained. The training process involves gradient descent optimization for token generation, which likely requires second-order gradient computation. However, this paper does not provide sufficient explanation of how second-order gradients are computed, their computational cost, or their impact on training efficiency.

- The attention mechanism in EBT may face efficiency challenges with second-order gradient computation. Modern LLMs and DiTs rely heavily on optimized implementations like Flash Attention for CUDA acceleration. However, these optimizations may not be compatible with second-order gradients, potentially limiting the computational efficiency of EBT's modified attention mechanism.

- The DiT experiments focus on image denoising, which is not representative of typical industrial applications. Most DiT usage in practice involves image generation or video generation tasks where denoising acceleration is highly demanded. This work would benefit from experiments on these more relevant applications to demonstrate EBT's effectiveness in real-world scenarios.

- Figure 5(a) and (b) show concerning patterns where EBT curves appear above the baseline in the upper right region. Given that the x-axis represents model size or compute and the y-axis represents perplexity, this suggests EBT achieves higher perplexity under similar conditions, which is not a positive trend. This may indicate a plotting issue.

**Questions:**

Please refer to the weaknesses section for specific technical questions. This work presents a novel approach to System 2 Thinking through Energy-Based Transformers, which provides a very novel perspective and contributes to the community. The work meets the acceptance bar for ICLR, but based on the noted issues, cannot receive an absolutely high score.

---

> ### Author Response · Authors · 2025-11-21
> **Response to Reviewer 8YzV (Part 1/2)**
>
> We are excited the reviewer states “EBTs address limitations of existing System 2 Thinking approaches”, that our work provides a very novel perspective, that the reviewer believes the paper is well written with comprehensive experimental design that demonstrates the robustness and effectiveness of EBTs, and thinks our core principle that verification is easier than generation is novel and insightful.
>
> > The verification approach in this work does not align with intuitive expectations. For language tasks, meaningful verification should operate on complete sentences or full responses to questions. However, EBT uses autoregressive generation where each token is verified independently during generation. This raises questions about the validity of verification at the token level rather than at the semantic level of complete thoughts or responses.
>
> **Response:** We agree that EBTs operating over an entire sequence would likely be better than operating over individual tokens. While experimenting with EBTs operating over entire sequences on data with many modes, i.e. the mentioned failed text-to-image experiments in Section B.3, we experienced poor generation quality/blurry images/mode collapse. We believe this is an issue with the current optimization formulation, which is unable to deal with a lot of uncertainty and plan to explore this issue as a promising future work.
>
> We briefly mentioned this challenge with capturing many modes in Future works under Section A.9, and have updated the text to highlight how important a direction this is. We have also added this as a limitation to be more explicit and upfront about it.
>
> We still believe that verification happening at the per-token level, such as with the AR-EBTs for language generation, is valuable due to enabling future-state verification with the given sequence. However, in the long-term, we agree with the reviewer’s sentiment regarding EBTs operating over entire sequences, and will work hard to realize this potential in future work.
>
>
> > This work lacks sufficient analysis of computational efficiency. The paper primarily uses forward pass counts as an indirect measure, but absolute metrics such as latency or other time-based measurements would provide a more objective assessment of EBT's computational performance compared to Transformer++. Such analysis is crucial for understanding the practical viability of EBT.
>
> **Response:** We kindly refer the reviewer to the [main comment](https://openreview.net/forum?id=ZBj3Qp1bYg&noteId=VbcSJxquF6), where we have conducted such experiments and added details to the main manuscript. We include an excerpt below:
>
> “In light of the reviewer’s helpful feedback to include discussions on computational overhead, we have added experiments and a more technical discussion in Section D.6. While the original manuscript contained information regarding theoretical FLOPs in Section D.5, we have added information regarding theoretical memory overhead as well as empirical wall clock time, memory overhead, and latency for EBTs vs. the Transformer++. Broadly, the empirical performances roughly match the theoretical FLOP/memory overheads predicted by our discussion in Section D.5.
>
> It’s worth mentioning that we leveraged PyTorch, which does not implement the necessary Hessian Vector Product calculations efficiently, and is far from being as optimized as Jax in this regard [6]. We strongly believe that algorithmic progress often drives system/software/hardware progress, and that if the field were to continue leveraging EBTs, that HVP computations could be sped up significantly so empirical performance better matches theoretical FLOPs/memory. Additionally, the recent advent of [new hardware specialized for EBMs](https://extropic.ai/writing/thermodynamic-computing-from-zero-to-one) offers promise in resolving many of the challenges with slower EBT computational efficiency.”
>
> We hope this clarifies the practical viability of EBT given current software/hardware.
>
> > The attention mechanism in EBT may face efficiency challenges with second-order gradient computation. Modern LLMs and DiTs rely heavily on optimized implementations like Flash Attention for CUDA acceleration. However, these optimizations may not be compatible with second-order gradients, potentially limiting the computational efficiency of EBT's modified attention mechanism.
>
> **Response:** We agree with the reviewer that there may be portions of EBTs that struggle to become computationally efficient due to the 2nd order derivatives required. We strongly believe that algorithms often drive progress in software/systems, such as with Transformers driving systems development with flash attention, and that if EBTs are widely adopted they may drive further algorithmic progress enabling efficient 2nd order computation. Fortunately, there has been recent progress in Flash Attention for 2nd order derivatives (HVPs), linked [here](https://github.com/amorehead/jvp_flash_attention).

---

> ### Author Response · Authors · 2025-11-21
> **Response to Reviewer 8YzV (Part 2/2)**
>
> > The implementation details of EBT are not clearly explained. The training process involves gradient descent optimization for token generation, which likely requires second-order gradient computation. However, this paper does not provide sufficient explanation of how second-order gradients are computed, their computational cost, or their impact on training efficiency.
>
> **Response:** We have updated the approach section to include more details regarding second-order gradient computation, including the following excerpt:
>
> “Importantly, this loss is backpropagated through the entire optimization process, requiring second-order derivatives (i.e., gradients of gradients). These are computed efficiently via Hessian-vector products, which scale linearly with model size [1].”
>
> We have also included details on how these second-order gradients are computed via Hessian-vector products in Section C.5, and note that more details on computational efficiency are in Sections D.5 and the newly added section D.6.
>
> [1] https://iclr-blogposts.github.io/2024/blog/bench-hvp/
>
> > The DiT experiments focus on image denoising, which is not representative of typical industrial applications. Most DiT usage in practice involves image generation or video generation tasks where denoising acceleration is highly demanded. This work would benefit from experiments on these more relevant applications to demonstrate EBT's effectiveness in real-world scenarios.
>
> **Response:** We agree that image and video generation are important industry use cases. While experimenting with EBTs for text-to-image generation, we observed poor generation quality due to the distribution's multimodality, which we discuss in Section B.3. We have recently added this challenge to the limitations section to be more upfront about it.
>
> At the same time, diffusion/DiT models are also widely used as policies in control and decision-making tasks [1–2]. A recent work has applied EBTs in this setting and directly compared EBT-based policies to diffusion-based ones [3], reporting stronger performance with substantially reduced inference-time computation. Although this study appeared after our submission, it provides external evidence that in more realistic, application-oriented scenarios, EBTs can match or even outperform diffusion models while preserving their computational benefits, consistent with the trends we observe in our denoising experiments.
>
> We have toned down claims regarding EBTs compared to Diffusion, and are happy to make any other changes requested.
>
> [1] https://arxiv.org/pdf/2303.04137
> [2] https://arxiv.org/pdf/2410.10088
> [3] https://arxiv.org/abs/2510.27545
>
> > Figure 5(a) and (b) show concerning patterns where EBT curves appear above the baseline in the upper right region. Given that the x-axis represents model size or compute and the y-axis represents perplexity, this suggests EBT achieves higher perplexity under similar conditions, which is not a positive trend. This may indicate a plotting issue.
>
> **Response:** Thanks for the clarification comments, you are correct that this suggests EBTs achieve higher perplexity, at the current scale EBTs are less computationally efficient than the Transformer++ baseline. However, the results also suggest that EBTs scale at an 8.97% higher rate, meaning that if the scaling laws continue to hold, EBTs may eventually become more compute-efficient than the Transformer++ recipe. While this has yet to be verified at scale, as we have limited computation, EBTs are a new approach and therefore there are likely many improvements that can be made to performance. It’s possible that future work making more architectural modifications could make EBTs more computationally efficient or even better on certain hardware [1].
>
> [1] https://extropic.ai/writing/thermodynamic-computing-from-zero-to-one

---

### Official Review · Reviewer_nhmE · 2025-10-30

**Soundness:** 3
**Presentation:** 3
**Contribution:** 3
**Rating:** 8
**Confidence:** 3

**Summary:**

This paper introduces Energy-Based Transformers (EBTs), a new framework that integrates energy-based modeling with Transformer architectures to enable what the authors describe as “System 2 Thinking.” Rather than directly predicting outputs, EBTs assign an energy value to input–output pairs, representing their compatibility, and perform inference through energy minimization. This allows the model to iteratively refine predictions and dynamically allocate computation based on task difficulty. The authors frame this as a form of unsupervised reasoning or self-verification. To make this scalable, they introduce several key techniques for stabilizing EBM training, including energy landscape regularization, Langevin noise, and replay buffers. EBTs are evaluated across language and vision domains, showing improved scaling behavior over standard Transformers, better generalization on out-of-distribution data, and significantly higher efficiency than Diffusion Transformers for denoising tasks.

**Strengths:**

The core contribution is genuinely novel and conceptually elegant. The insight that verification is easier than generation (grounded in complexity theory) is well-motivated, and coupling the verifier and generator through energy gradients avoids adversarial dynamics. The cross-modal validation across language, video, and images demonstrates generality beyond domain-specific tricks. Most compellingly, Figure 6 showing that thinking gains increase with distributional shift mirrors human cognition and suggests the approach addresses fundamental generalization challenges. The experimental analysis is thorough with good ablations and comprehensive scaling experiments across six axes.The technical execution is solid, with practical solutions to known EBM training challenges (replay buffers, Langevin dynamics, path randomization) and honest discussion of limitations including text-to-image generation failures. The OOD generalization results showing better downstream performance despite worse pretraining perplexity are particularly interesting. The extensive implementation details and promised code support reproducibility, and the work opens important research directions around reasoning without external supervision and uncertainty quantification in continuous spaces.

**Weaknesses:**

The scale limitations severely undermine the paper's claims. All experiments max out at 800M parameters while making assertions about foundation model behavior through extrapolation. The paper uses extrapolation to larger sized for many claims, which is speculative given that scaling laws often break at different regimes. Without validation the central claims about foundation model potential remain unsubstantiated.
The computational costs (3.33-6.66× FLOPs for training, gradient computation overhead at inference) are dismissed too casually as a temporary inconvenience rather than a fundamental barrier. At foundation model scale, this multiplier translates to enormous real-world costs. The paper lacks wall-clock time comparisons, memory analysis, or ablations showing what standard transformers could achieve with the same computational budget. The "System 2 Thinking" here is essentially inference-time optimization and the analogy to human reasoning is somewhat overstated, as it doesn't incorporate concrete verbose reasoning.
Critical baselines are missing, particularly comparisons to recent inference-time scaling methods. The Transformer++ baseline showing "no improvement" with more compute is somewhat unfair since it wasn't designed for that.

**Questions:**

1. You report that EBTs require 3.33-6.66× more FLOPs than standard Transformers during training. However, the paper doesn't compare what standard Transformers could achieve with an equivalent computational budget (e.g., training with 3× more parameters, 2× depth, or simply training longer). Could you provide an ablation where Transformer++ is given the same total compute budget as EBT? This would help disentangle whether improvements come from the EBM formulation itself or simply from using more computation. Additionally, what are the wall-clock training times and memory requirements compared to Transformer++ on identical hardware?
2. Section B.3 says that EBTs fail on text-to-image generation with multiple modes, producing blurred images. This seems fundamental, yet the paper broadly claims superior generalization for EBTs. Can you clarify: (a) What classes of problems will inherently fail with EBTs due to such limitations? (b) Is there a fundamental trade-off between the strong OOD generalization you demonstrate and the ability to handle multimodal output distributions? (c) For language modeling specifically, how does this limitation manifest (e.g. does it affect creative generation tasks that benefit from diversity)?

---

> ### Author Response · Authors · 2025-11-21
> **Response to Reviewer nhmE (Part 1/3)**
>
> We thank the reviewer for the great suggestions. We are happy the reviewer finds the core contribution genuinely novel and conceptually elegant, that the experiments suggest generality and are comprehensive with good ablations, and that the experiments suggest our approach addresses fundamental generalization challenges (Figure 6).
>
> > The scale limitations severely undermine the paper's claims. All experiments max out at 800M parameters while making assertions about foundation model behavior through extrapolation. The paper uses extrapolation to larger sized for many claims, which is speculative given that scaling laws often break at different regimes. Without validation the central claims about foundation model potential remain unsubstantiated.
>
> **Response:** We thank the reviewer for this insightful feedback. We have updated the manuscript to revise and reduce extrapolative and unsubstantiated claims, and are happy to to update any other specific locations.
>
> While we agree that 800M parameters is not at the size of recent foundation models, we are constrained by computing resources from going beyond such model sizes. To reduce uncertainty regarding the scalability of EBTs, as detailed in the [general response](https://openreview.net/forum?id=ZBj3Qp1bYg&noteId=VbcSJxquF6), we have reconducted the primary scaling law experiments (Figures 5a and 5b) to improve the reliability and strength of these experiments, including multiple seeds as well as more robust hyperparameters for scaling data and parameters (see the main rebuttal comment for details). The results when reconducting these scaling laws are that EBTs have a scaling rate that is 8.97% higher than the Transformer++ in language modeling. These results are more significant than our earlier results. While they don’t demonstrate that EBTs scale beyond 800M params, they provide stronger evidence that EBTs scale at a higher rate than the Transformer++.
>
> >The computational costs (3.33-6.66× FLOPs for training, gradient computation overhead at inference) are dismissed too casually as a temporary inconvenience rather than a fundamental barrier. At foundation model scale, this multiplier translates to enormous real-world costs. The paper lacks wall-clock time comparisons, memory analysis, or ablations showing what standard transformers could achieve with the same computational budget.
>
> >You report that EBTs require 3.33-6.66× more FLOPs than standard Transformers during training. However, the paper doesn't compare what standard Transformers could achieve with an equivalent computational budget (e.g., training with 3× more parameters, 2× depth, or simply training longer). […] Additionally, what are the wall-clock training times and memory requirements compared to Transformer++ on identical hardware?
>
> **Response:** Thanks for the great feedback regarding the inclusion of empirical computational costs (including memory, wall-clock time, etc). We have added such results in Section D, which includes empirical results on wall clock time, memory overhead, and latency for EBTs vs. the Transformer++. The [main rebuttal](https://openreview.net/forum?id=ZBj3Qp1bYg&noteId=VbcSJxquF6) includes a discussion of these costs in more detail, which we include below:
>
> “In light of the reviewer’s helpful feedback to include discussions on computational overhead, we have added experiments and a more technical discussion in Section D.6. While the original manuscript contained information regarding theoretical FLOPs in Section D.5, we have added information regarding theoretical memory overhead as well as empirical wall clock time, memory overhead, and latency for EBTs vs. the Transformer++. Broadly, the empirical performances roughly match the theoretical FLOP/memory overheads predicted by our discussion in Section D.5.
>
> It’s worth mentioning that we leveraged PyTorch, which does not implement the necessary Hessian Vector Product calculations efficiently, and is far from being as optimized as Jax in this regard [6]. We strongly believe that algorithmic progress often drives system/software/hardware progress, and that if the field were to continue leveraging EBTs, that HVP computations could be sped up significantly so empirical performance better matches theoretical FLOPs/memory. Additionally, the recent advent of [new hardware specialized for EBMs](https://extropic.ai/writing/thermodynamic-computing-from-zero-to-one) offers promise in resolving many of the challenges with slower EBT computational efficiency.”
>
> We have added more discussion in the limitations section of the paper regarding computational costs as a foundational barrier to overcome–we hope this makes it clearer how much of a challenge it’ll be to integrate EBTs due to them requiring more computation/memory.

---

> ### Author Response · Authors · 2025-11-21
> **Response to Reviewer nhmE (Part 2/3)**
>
> Additionally, other approaches for achieving strong reasoning, such as RL, are also computationally intensive. RL rollouts often involve 10-1000x the amount of computational time as normal pre-training for fewer bits of supervision. Our results can therefore be viewed as exploring a similar trade-off: EBTs convert additional FLOPs and memory into improved generalization/reasoning. Therefore, as the field continues to progress towards investing more computation for improved generalization and reasoning, we believe EBTs are a promising approach.
>
> > Could you provide an ablation where Transformer++ is given the same total compute budget as EBT? This would help disentangle whether improvements come from the EBM formulation itself or simply from using more computation.
>
> **Response:** Thanks for the suggestion regarding comparing Transformers with an equivalent computational budget to EBTs. We are currently conducting more experiments to investigate this, but we would like to highlight that Figure 5b shows what the Transformer++ can achieve with the same compute budget as EBTs. At the current scale, the Transformer++ outperforms EBT significantly, as EBTs are more compute-intensive. However, as the scaling rate of EBTs is higher by 8.97%, this means that at much larger scale EBTs may eventually become more compute efficient than the Transformer++, although this is speculative as these experiments are prohibitively expensive. We are happy to conduct any experiments beyond what Figure 5b shows if the reviewer could elaborate in more detail.
>
> > The "System 2 Thinking" here is essentially inference-time optimization and the analogy to human reasoning is somewhat overstated, as it doesn't incorporate concrete verbose reasoning.
>
> **Response:** We appreciate this feedback and agree that our use of “System 2 Thinking” primarily encompasses inference-time optimization, and that the analogy to human reasoning may be somewhat overstated given the lack of explicit, verbose reasoning. We discuss the nuances of this terminology in Section G, including why we view “System 2 Thinking” as a useful and general umbrella term, and in what sense current approaches remain far from true human-like System 2 Thinking (though we see EBTs as a step in the right direction).
>
> The requirement for concrete verbose reasoning (e.g., Chain-of-Thought) [1] assumes that human System 2 Thinking operates through discrete tokens or language, but it is not definitively clear whether human reasoning is fundamentally based on this discrete, verbal mode, or whether it also involves iterative refinement and verification in a more continuous, latent space [2]. For instance, many animals are able to reason without discrete language [3]. EBTs explore the latter perspective by modeling thinking as a continuous optimization procedure with respect to a learned verifier, which we view as a promising avenue toward a more generalized and modality-agnostic form of reasoning emerging from unsupervised learning. We have updated the main text to more clearly point to this discussion in Section G.
>
> [1] https://arxiv.org/abs/2201.11903
> [2] https://arxiv.org/pdf/2412.06769
> [3] https://www.science.org/doi/10.1126/science.1098410
>
> > Critical baselines are missing, particularly comparisons to recent inference-time scaling methods.
>
> **Response:** We have added experiments with recurrent neural networks (RNNs) to add a more commonly used inference-time scaling method. We did this for the recently conducted data-constrained Sudoku experiments, which have been added in the Experimentation Section. We compared a standard feed-forward Transformer, an RNN Transformer (similar to [1-2]), and EBTs. Our results (described in the experimentation Section) are strong with EBTs scoring 29.7% on the out-of-distribution test set, RNNs scoring 17.7% on the test set, and standard feed-forward Transformers scoring 0.03%--suggesting EBTs can generalize better due to their reasoning capabilities. These results are consistent with prior work on EBMs in data-constrained reasoning tasks [3-4].
>
> | Model         | OOD Test-Set Performance (%) |
> | ------------- | ---------------------------- |
> | Transformer++ | 0.03                          |
> | RNN           | 17.7                         |
> | EBT           | 29.7                         |
>
>
> ​​[1] https://arxiv.org/abs/2510.04871
> [2] https://arxiv.org/abs/1807.03819
> [3] https://arxiv.org/abs/2406.11179
> [4] https://arxiv.org/pdf/2206.15448

---

> ### Author Response · Authors · 2025-11-21
> **Response to Reviewer nhmE (Part 3/3)**
>
> > The Transformer++ baseline showing "no improvement" with more compute is somewhat unfair since it wasn't designed for that.
>
> **Response:** We have clarified that the Transformer++ is not designed for this type of experiment in the caption of Figure 7a. We hope that this makes Figure 7a clearer.
>
> > Section B.3 says that EBTs fail on text-to-image generation with multiple modes, producing blurred images. This seems fundamental, yet the paper broadly claims superior generalization for EBTs. Can you clarify: (a) What classes of problems will inherently fail with EBTs due to such limitations? (b) Is there a fundamental trade-off between the strong OOD generalization you demonstrate and the ability to handle multimodal output distributions? (c) For language modeling specifically, how does this limitation manifest (e.g. does it affect creative generation tasks that benefit from diversity)?
>
> **Response:** Thanks for the interesting questions, we answer them below:
>
> (a) EBTs currently fail in classes of problems with many modes that need to be modeled, given limited conditioning, such as generating an image from text or unconditional video generation. If we perform a task such as autoregressive language/video modeling, the number of modes decreases, and hence EBTs are able to succeed. We believe this is an issue with the current optimization formulation and plan to explore it as a promising future work.
>
> (b) We have not observed any trade-off between strong OOD generalization and the ability to handle multimodal output distributions from our experiments as well as related works [1-2], and believe that it is possible to obtain both strong OOD generalization and the ability to handle multimodal output distributions with the right training objective.
>
> (c) For autoregressive language modeling, we have not observed any issues with this limitation, as the space is discrete and hence modeling the modes (i.e., the next token autoregressively) is easier than in continuous spaces. Consequently, as we have observed improved generalization with EBTs, it’s possible that EBTs actually help with tasks such as creativity that benefit from diversity.
>
> We have added more discussion of these challenges in the limitations and future work sections.
>
> [1] https://arxiv.org/abs/2302.11552
> [2] https://arxiv.org/abs/2406.11179

---

> > ### Comment · Reviewer_nhmE · 2025-11-27
> >
> > Thank you for your detailed response. I understand that some of my concerns cannot be addressed due to compute constraints, however I suggest a more careful framing in the paper to prevent over-confident predictions about the scalability of findings to foundation models. Overall my assessment of the paper, and my recommendation for acceptance, have been reinforced and I will maintain my score.

---

> ### Author Response · Authors · 2025-12-03
>
> We thank the reviewer for their continued recommendation of acceptance for the paper. We have modified the paper to discuss key results and findings more carefully, and make it more explicit that scaling performance to the size of modern foundation models remains untested/unknown. We thank the reviewer for this suggestion.

---

### Official Review · Reviewer_k52K · 2025-11-04

**Soundness:** 3
**Presentation:** 2
**Contribution:** 3
**Rating:** 6
**Confidence:** 3

**Summary:**

This paper proposes a novel framework of an energy-based transformer model. This model integrates a new method that allows efficient and scalable training of EBMs, and thus allows inference-time verification.

**Strengths:**

- I think the core idea, integrating self-verification in inference time, is promising.
- The empirical results verify the effectiveness of the proposed model.

**Weaknesses:**

- Although this paper is overall clear, I think the authors have spent too much space introducing related concepts, instead of the technical details of the proposed method. It is unclear how the model specifically works. It would be good if there could be an extra section to explain the details of the model, perhaps through a toy example.
- It is unclear why "frame EBM as an optimization problem" can avoid the curse of dimensionality. All evidences provided in the paper are just vague discussions, and I don't see any formal analysis. Also, it is unclear how this "frame EBM as an optimization problem" actually works.

**Questions:**

N/A

---

> ### Author Response · Authors · 2025-11-21
> **Response to Reviewer k52k (Part 1/1)**
>
> We thank the reviewer for their valuable feedback. We appreciate that the reviewer thinks that the empirical results verify the effectiveness, the method allows for efficient and scalable training of EBMs, the core idea is promising, and the idea enables inference-time verification.
>
> >Although this paper is overall clear, I think the authors have spent too much space introducing related concepts, instead of the technical details of the proposed method. It is unclear how the model specifically works. It would be good if there could be an extra section to explain the details of the model, perhaps through a toy example.
>
> **Response:** We thoroughly appreciate this feedback and have updated the manuscript approach section to include more technical details on the proposed method, with the following excerpts:
>
> “Intuitively, this optimization-based training approach is similar to GANs [1]. During the forward pass, EBMs can be seen as a GAN discriminator by giving an energy "verification"; on the backward pass, they can be seen as a GAN generator by optimizing predictions through energy minimization to try and fool the discriminator.”
>
> “Then, the loss can be computed using any standard objective function.
> Importantly, this loss is backpropagated through the entire optimization process, requiring second-order derivatives (i.e., gradients of gradients). These are computed efficiently via Hessian-vector products, which scale linearly with model size, similar to standard gradient descent in feed-forward models. More details and pseudocode can be found in Section I.2.”
>
> The idea to include a toy example is excellent, and we have included the following example in **Section C:**
>
> “Similar to standard neural network training, the optimization process for EBTs starts from an initial random guess. For neural networks, this corresponds to randomly initialized weights; in our EBT formulation, it corresponds to initializing the prediction $y_0$ by sampling from a Gaussian distribution. In neural networks, we compute a loss and differentiate it to obtain a gradient with respect to the network parameters, which we then update to reduce the loss.
>
> In contrast, our method directly optimizes the prediction. Given a current prediction $y_t$, we input $y_t$ (and the conditioning $x$, if applicable) into the EBM to obtain a scalar energy $E_\theta(x, y_t)$. We then compute the gradient of this energy with respect to the prediction, $\nabla_{y} E_\theta(x, y_t)$, and update the prediction via gradient descent to decrease the energy. This iterative procedure for updating the prediction mirrors the way neural network parameters are updated during training to minimize the loss, but operates in prediction space instead of parameter space.”
>
> We would like to highlight that the original manuscript included technical details in Algorithm 1 and 2, Equations 1 and 2, as well as Listings 1 and 2 in the appendix with pseudocode for implementations. We are happy to provide any other technical details, and hope that these clarify how the model works.
>
> [1] https://arxiv.org/abs/1406.2661
>
> >It is unclear why "frame EBM as an optimization problem" can avoid the curse of dimensionality. All evidences provided in the paper are just vague discussions, and I don't see any formal analysis. Also, it is unclear how this "frame EBM as an optimization problem" actually works.
>
> **Response:** To make it clearer how framing EBM training as an optimization problem can avoid the curse of dimensionality, we have described this in Section C, with the following text:
>
> “This approach avoids the curse of dimensionality inherent in traditional contrastive Energy-Based Model (EBM) training by reframing learning as an optimization problem (Algorithm 1). Contrastive methods must explicitly push up the energy of an exponentially increasing number of negative samples in high-dimensional space. Our optimization-based approach, instead, implicitly shapes the energy landscape by training the network to successfully perform gradient descent from a random initial prediction to the ground truth solution. This implicitly regularizes the landscape to only have local energy minima at data points. Because regularized methods are known to suffer less from the curse of dimensionality, this successfully avoids such a pitfall [1]. This aligns with more formal results on Ising models from theoretical computer science, where learning the energy function (Hamiltonian) is of much simpler complexity-wise than accurately sampling or learn the full resulting probability distribution [2].”
>
> We hope that the added/included technical details linked above, including the toy example, the algorithms, listings and equations help to improve clarity regarding the EBT optimization formulation, although we are happy to provide any other details the reviewer finds necessary.
>
> [1] https://openreview.net/pdf?id=BZ5a1r-kVsf
> [2] https://www.jmlr.org/papers/v11/ravikumar10a.html

---

### Author Response · Authors · 2025-11-21
**General Response (Part 1/2)**

We thank the reviewers for their great feedback and suggestions regarding the paper. We are happy the reviewers find the core idea genuinely novel and/or elegant (Reviewers k52K, nhmE, 8YzV, KJ9J), and that the empirical results broadly demonstrate the effectiveness of the proposed Energy-Based Transformers (EBTs) (Reviewers k52K, nhmE, 8YzV).

The primary points raised relate to improving scaling experiments and clarifying evidence/details, adding key experimental comparisons, detailing computational overhead, and adding implementation specifics.

First, motivated by the reviewer’s insightful feedback regarding limited experiments at scale with limited evidence for scaling claims, we conducted a new set of primary scaling law experiments (Figures 5a and 5b), with revised scaling parameters to improve the reliability and strength of these results. Although the initial manuscript included details on these experiments in Section D.1.1, we have updated the manuscript to provide more comprehensive details and have updated Figures 5a and 5b with the latest results. For these experiments, we have decided to use scaling parameters more aligned with the literature, where we follow Chinchilla [1] to scale model sizes and data proportionally (earlier experiments did not scale model sizes and data proportionally). We have rerun these experiments with multiple random seeds, computed an $R^2$ value to determine significance, and have reported tables with our results in Section D.1.1, where we find that EBTs have a scaling rate 8.97% higher than the Transformer++ for language modeling (which is higher than before, with 2.92%). While we are limited by computation from conducting much larger-scale experiments (e.g. billions of parameters), we believe that results holding across seeds and more common scaling parameters substantiate our results further. We hope that these experiments have addressed comments regarding scalability and thoroughness of scaling law experiments, and we have updated claims to better reflect that we are unsure what large-scale performance will be with billions of parameters.

The reviewers offered several experimental suggestions to conduct, including data-constrained experiments, other inference-time compute models (i.e., RNNs), and empirical overhead experiments. We believe we have broadly conducted and added all of these experiments to the manuscript.

For the data-constrained environments, we chose to use sudoku, as suggested by Reviewer KJ9J, and compared a standard feed-forward Transformer, an RNN Transformer (similar to [2-3]), and EBTs. Our results, described in the Experimentation Section, are strong, with EBTs scoring 29.7% on the out-of-distribution test set, RNNs scoring 17.7%, and standard feed-forward Transformers scoring 0.03%--suggesting EBTs can generalize better. These results are in line with existing results exploring EBMs in data-constrained reasoning tasks [4-5].

| Model         | OOD Test-Set Performance (%) |
| ------------- | ---------------------------- |
| Transformer++ | 0.03                          |
| RNN           | 17.7                         |
| EBT           | **29.7**                         |

In light of the reviewer’s helpful feedback to include discussions on computational overhead, we have added experiments and a more technical discussion in Section D.6. While the original manuscript contained information regarding theoretical FLOPs in Section D.5, we have added information regarding theoretical memory overhead as well as empirical wall clock time, memory overhead, and latency for EBTs vs. the Transformer++. Broadly, the empirical performances roughly match the theoretical FLOP/memory overheads outlined in our discussion in Section D.5.

It’s worth mentioning that we leveraged PyTorch, which does not implement the necessary Hessian Vector Product calculations efficiently, and is far from being as optimized as Jax in this regard [6]. We strongly believe that algorithmic progress often drives system/software/hardware progress, and that if the field were to continue leveraging EBTs, that HVP computations could be sped up significantly so empirical performance better matches theoretical FLOPs/memory. Additionally, the recent advent of [new hardware specialized for EBMs](https://extropic.ai/writing/thermodynamic-computing-from-zero-to-one) offers promise in resolving many of the challenges with slower EBT computational efficiency.

Fourth, based on reviewer suggestions, we have clarified several implementation details of EBTs in the approach section. The original manuscript contained implementation details in Algorithms 1 and 2, Equations 1 and 2, as well as Listings 1 and 2 in the appendix with pseudocode for implementations. To add to these implementation details, in Section C, we have added a toy example of how the model functions, an explanation of avoiding the curse of dimensionality, and how computing second-order gradients works.

---

> ### Author Response · Authors · 2025-11-21
> **General Response (Part 2/2)**
>
> ​​[1] https://arxiv.org/pdf/2203.15556
> [2] https://arxiv.org/abs/2510.04871
> [3] https://arxiv.org/abs/1807.03819
> [4] https://arxiv.org/abs/2406.11179
> [5] https://arxiv.org/pdf/2206.15448
> [6] https://iclr-blogposts.github.io/2024/blog/bench-hvp

---

### Author Response · Authors · 2025-12-04
**Response to AC After Discussion Phase (Part 1/2)**

Dear Area Chair,

Thank you for your time and effort in reviewing our submission. Following the recommendation of the ICLR 2026 Program Chairs, we would like to provide a brief summary of the discussion phase for this paper.

**Reviewer KJ9J:**
- Reviewer KJ9J mentioned that our approach enables two appealing properties associated with System 2 reasoning, our results exhibit competitive performance across discrete and continuous domains as well as under distribution shifts, the core idea feels conceptually elegant, and the results suggest EBTs offer promise for inference-time reasoning that emerges from pretraining.
- Reviewer KJ9J noted limitations primarily regarding our scaling law experiments, shown in Figures 5a and 5b. We took care to reconduct these experiments in our [main rebuttal response](https://openreview.net/forum?id=ZBj3Qp1bYg&noteId=VbcSJxquF6) more thoroughly, using multiple random seeds, aligning the experimental setup more with the literature (following Chinchilla scaling better), and calculating $R^2$ values for determining the significance of results. These newly conducted results suggest that **EBTs have a scaling rate 8.97% higher than the Transformer++** for language modeling (which is higher than before, with 2.92%).
- Reviewer KJ9J also suggested running experiments in data-constrained settings, which we conducted using Sudoku benchmarks. Our results demonstrate that EBTs consistently out-generalize existing models, with **EBTs scoring 29.7%** on the out-of-distribution test set, **RNNs scoring 17.7%,** and standard **feed-forward Transformers scoring 0.03%.**
With these changes to the paper and new results, Reviewer KJ9J **increased their score from a 2 to a 6** on November 23rd, as mentioned in [this rebuttal comment](https://openreview.net/forum?id=ZBj3Qp1bYg&noteId=qAMswUGpXz), describing how their “concerns are addressed” and how they “look forward to learning more about this paper at ICLR.”

**Reviewer k52K:**
- Reviewer k52K, with an initial score of 6, found the framework to be novel, that the approach enables efficient and scalable training of EBMs, and that the empirical results verify the effectiveness of the approach.
- Reviewer k52K raised points regarding a lack of technical details, clarity regarding the optimization-based EBM formulation, and suggested incorporating a toy example. We have added several technical details, including an additional half page of content in the approach section describing training and inference algorithms, the toy example demonstrating how EBTs work, and 2 additional subsections in Section C with technical details.
- Unfortunately, Reviewer k52K never had the opportunity to respond, however, we believe we addressed all points through adding several additional technical details/examples/discussions on how EBTs work in the approach section and in Section C.

**Reviewer nhmE**
- Reviewer nhmE, with an initial score of 8, found EBTs “genuinely novel and conceptually elegant”, found the experiments comprehensive and thorough, and stated that the experiments suggest EBTs address fundamental generalization challenges.
- Reviewer nhmE requested a more rigorous dive into empirical efficiency (i.e. wall clock time, memory, and latency), FLOP-matched experiments, and insight into modeling multimodal distributions. We conducted experiments measuring empirical efficiency and added these into Section D.6, discussed our FLOP-matched experiments in Figure 5b, and added more of a discussion on the promising potential of EBTs for modeling multimodal distributions in the limitations and broader impact sections.
- Reviewer nhmE responded to our discussion/rebuttal, noting “my assessment of the paper, and my recommendation for acceptance, have been reinforced and I will maintain my score.”


**Reviewer 8YzV**
- Reviewer 8YzV, giving an initial score of 8, stated that our work provides a very novel perspective, that the paper is well written with a comprehensive experimental design that demonstrates the robustness and effectiveness of EBTs, and stated our core principle that verification is easier than generation is novel and insightful.
- Reviewer 8YzV recommended more experiments regarding empirical efficiency, discussion of verification at the sequence level, and technical details on the approach. We conducted comprehensive experiments for measuring efficiency empirically and added these into Section D.6, added a discussion of EBTs for verification at the sequence level and associated challenges in the limitations and broader impact sections, and added more technical details in the main paper as well as in the appendix with 3 additional Subsections in Section C
- Unfortunately, 8YzV was unable to respond to our discussions. However, we believe we addressed all concerns via conducting more experiments, adding more technical details, and adding more discussions on current limitations, among other changes.

---

> ### Author Response · Authors · 2025-12-04
> **Response to AC After Discussion Phase (Part 2/2)**
>
> We would like to highlight that all reviewers found the core idea genuinely novel and/or elegant, and all reviewers found that the results demonstrate the effectiveness/competitiveness of the proposed Energy-Based Transformers (EBTs).
>
> We hope this summary provides a clear overview of how discussions unfolded and how the paper was updated accordingly. We would be happy to offer any additional clarification if needed, and we thank the AC for their time.
>
> Sincerely,
> Authors of Submission 14452

---

### Meta-Review · Area_Chair_jYpX · 2025-12-22

**Summary:**

(*Disclaimer: given the peculiar review process, some of my choices and reasonings below will be highly subjective, as I tried to imagine how a reviewer would have reacted to a specific response. I understand that any negative choice will be perceived as unfair by the authors, and I apologize in advance for that.*)

(*Second disclaimer: the authors and some reviewers explicitly mention some changes in scores that occurred during the rebuttal. As these were reverted due to the possibility of collusion in light of the security incident, I will tend to disregard this information.*)

The paper proposes an energy-based model as alternative to existing autoregressive and diffusion models. The core idea is to define an energy function over pairs of prompts and next tokens, and provide a prediction as a few-shot gradient descent on the energy landscape. They provide substantial experiments showing interesting results on "system 2" thinking (i.e., allocating more computation during inference). However, the paper remains a proof-of-concept due to (a) small size of tested models (up to 800M parameters), (b) a scaling law analysis requiring 2-3 orders of magnitude more computation before the method becomes more viable than alternatives, (c) non-optimized implementation.

The paper received four reviews, which included 2 strong acceptances, 1 strong rejection (`KJ9J`), and 1 extremely shallow review (`k52K`) voting for a borderline acceptance. Focusing on the three core reviewers, they all agree the method is potentially interesting, the paper is extremely well written and motivated, and the formulation is elegant and clear.

The main concerns revolved around theoretical and practical cost. In response, the authors provided a new scaling law analysis, more details on the implementation, and they also revised or toned down multiple claims in the paper. Overall, as detailed below, reviewers who participated in the rebuttal were satisfied, and it is probable the paper would have ended the rebuttal with a consensus towards a strong acceptance.

Summarizing, this is a paper that proposes a promising line of research with intriguing results. The rebuttal was interesting and the paper was significantly strengthened by the additions. Thus, the paper meets the bar for ICLR acceptance by a strong margin despite a lack of state-of-the-art results.

**Reviewer Concerns:**

(*I will focus on some key weaknesses identified by multiple reviewers.*)

**Scaling law analysis** (`KJ9J`, `nhmE`): the authors use a scaling law analysis to argue that in the future the method would overcome alternative approaches given its better slope. However, the original analysis had unrealistic exponents. The authors completely revamped the analysis in the rebuttal, and the new results are more reasonable (albeit not a "proof" in any real sense).

**Efficiency** (`KJ9J`, `8YzV`, `nhmE`): among other things, the method requires second-order derivatives for back-propagation and standard derivatives for inference. The authors have added experiments with wall-clock time and some clarifications on how HVPs are implemented. They also point to multiple future improvements (e.g., JAX, EMB-friendly hardware).

**Multiple hyper-parameters** (`KJ9J`): the authors argue this is a "*feature, not a bug*" as it provides more freedom to the user. There was no further discussion on this point.

**Additional experiments** (`KJ9J`, `8YzV`, `nhmE`): this includes experiments against iterative (e.g., looped) baselines, data-constrained settings, and generation tasks. The authors have added a very interesting set of experiments on Sudoku puzzle answering (in part) the first two points, as well as concurrent works investigating the use of EBMs for generation.

**Additional details on the method** (`nhmE`, `k52K`): the authors added some clarifications in the manuscript, although there was no further discussion in the rebuttal.

**Reviewer Scores:**

`KJ9J`: while the reviewer originally voted for a strong rejection, they also made a very positive evaluation on the paper's potential. During the rebuttal, several questions were answered (while several were left for future work) and the reviewer was satisfied. It was probable they would have ended the rebuttal with a (at least) weak acceptance score.

`8YzV`: the reviewer voted for a strong acceptance and did not answer during the rebuttal.

`nhmE`: also voted for strong acceptance and was satisfied by the rebuttal.

`k52K`: the review was very shallow and the reviewer never participated in the rebuttal.

---

### Decision · Program_Chairs · 2026-01-26

Accept (Oral)